# SocioDojo: Building Lifelong Analytical Agents with Real-world Text and Time Series

**Junyan Cheng**
Thayer School of Engineering
Dartmouth College
Hanover, NH 03755, USA
`jc.th@dartmouth.edu`

**Peter Chin**
Thayer School of Engineering
Dartmouth College
Hanover, NH 03755, USA
`pc@dartmouth.edu`

## ABSTRACT

We introduce SocioDojo, an open-ended lifelong learning environment for developing ready-to-deploy autonomous agents capable of performing human-like analysis and decision-making on societal topics such as economics, finance, politics, and culture. It consists of (1) information sources from news, social media, reports, etc., (2) a knowledge base built from books, journals, and encyclopedias, plus a toolbox of Internet and knowledge graph search interfaces, (3) 30K high-quality time series in finance, economy, society, and polls, which support a novel task called "hyperportfolio", that can reliably and scalably evaluate societal analysis and decision-making power of agents, inspired by portfolio optimization with time series as assets to "invest". We also propose a novel Analyst-Assistant-Actuator architecture for the hyperportfolio task, and a Hypothesis & Proof prompting for producing in-depth analyses on input news, articles, etc. to assist decision-making. We perform experiments and ablation studies to explore the factors that impact performance. The results show that our proposed method achieves improvements of 32.4% and 30.4% compared to the state-of-the-art method in the two experimental settings.

## 1 INTRODUCTION

Autonomous agents based on large language models (LLMs) have made considerable progress in task-oriented settings, such as software development (Qian et al., 2023), web navigation (Zhou et al., 2023), and robot planning (Chen & Chang, 2023). These tasks usually require only task-specific skills rather than open-ended knowledge about human society. For instance, to make economic and financial decisions, one must keep track of the latest events and trends as well as have a broad understanding of how society works. Such scenarios are ubiquitous in the human world from personal finance and business planning to social science studies and public affairs. Therefore, it is essential to develop agents that can comprehend, analyze, and make decisions in an open-ended human society to achieve human-centered AI.

Human society is built on information. We create **SocioDojo**, a lifelong learning environment that emulates the historical development of society with real-world texts and time series. SocioDojo has three components: information sources, time series, and a knowledge base & toolbox. It uses a continuous stream of messages from multiple sources, such as news, articles, etc., to simulate the dynamics of society. Agents are constantly updated with the latest messages to keep track of changes in society. Time series from finance, economics, search trends, and poll tracking are used to depict the inner states of society. The knowledge base, made up of books, journals, and encyclopedias, along with a toolbox that provides Internet access, gives the necessary background knowledge. The use of real data also allows SocioDojo to automatically extend and the agents developed on it to be transferred into real applications without any gaps.

A scalable way to assess the societal analysis and decision-making capabilities of agents is needed, as the popular question-answering approach is expensive in generating up-to-date and high-quality questions that keep up with societal changes. We introduce the **hyperportfolio** task, which evaluates the agent through their "investment" decisions over time, which requires a thorough understanding

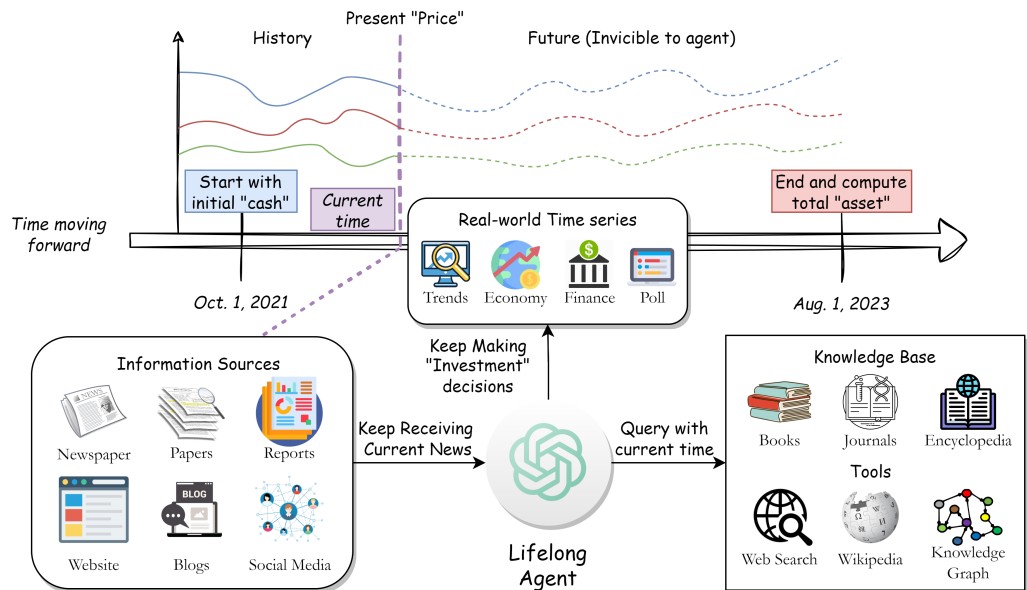

Figure 1: Illustration of the hyperportfolio task in SocioDojo. The agent begin with an initial cash of one million on October 1, 2021. Messages from information sources will be continually popping out to the agent, with the current time set as the published time of the newly popped message. The agent can "invest" the "assets" that correspond to the time series which "price" is the current value in the time series, based on the message, knowledge base, and tools. The goal is to maximize the total asset by the end time, August 1, 2023.

of society and sophisticated analysis even for human experts. We extend investable assets from financial assets to all kinds of time series in SocioDojo. An illustration of the hyperportfolio task is shown in Figure 1.

SocioDojo presents three unique open challenges. (1) **Spatiotemporal and entity information.** The time, place, and entity involved in an event are essential. Agents may need to link clues from multiple times, locations, and entities to reach a conclusion. (2) **Expert knowledge in the open domain.** To make an analysis or a decision about society, agents may require expert knowledge in multiple undetermined domains. (3) **Complex and hybrid reasoning.** Agents need both logical and numerical reasoning and the use of tools, including even machine learning models, to perform in-depth analysis. Addressing these challenges also facilitates the exploration of general intelligence.

We also present a novel Analyst-Assistant-Actuator architecture and Hypothesis-Proof prompting for the hyperportfolio task, which improves the state-of-the-art method by 32.4% and 30.4% in two settings in our experiment. We also conduct ablation studies on information sources and foundation models. Our key contributions are as follows.

- We create SocioDojo with Hyperportfolio task, the first large-scale and easily scalable environment built with real-world texts, time series, and search interfaces, enabling development and reliable assessment of the analytical and decision-making agents for real society;
- We propose Analyst-Assistant-Actuator architecture and Hypothesis-Proof prompting for producing in-depth analyses supporting high-quality decision-making;
- We perform experiments and ablation studies to explore factors that impact performance.

Our code and data are available at `https://github.com/chengjunyan1/SocioDojo`.

## 2 RELATED WORK

**Agent Learning Environment.** Most learning environments focus on the execution of specific tasks. For example, Zhou et al. (2023); Yao et al. (2023a) created simulated environments to develop

web agents that perform web browsing tasks. Toyama et al. (2021) proposed the use of an Android simulator to learn phone operations. Puig et al. (2018); Shridhar et al. (2020; 2021) aimed to train embodied agents to reason in a home environment. Non-task-specific open-ended agents have been researched through video games. Fan et al. (2022); Johnson et al. (2016); Shah et al. (2021) proposed the use of the open-world MineCraft game to study a skill set of exploration, planning, etc. for agents, while Vinyals et al. (2017) attempted to use the real-time strategic game StarCraft II to promote strategic thinking and multi-agent planning. Simulated human societies have been studied in *empire-building games*, with Branavan et al. (2011) developed agents in the Civilization II environment. However, the analytical skills learned in games largely deviate from real society.

**Reasoning in Large Language Model Agent.** Reasoning is a fundamental skill for agents performing analysis. Wei et al. (2022) encourages the model to reason step by step. Yao et al. (2023b) further introduces tree search into the reasoning process. Wang et al. (2023b) suggests using multiple reasoning paths instead of a single greedy approach to ensure better self-consistency. However, these approaches assume that the analysis process can be broken down into atomic steps, which may not be applicable to open-ended problems. Yao et al. (2023c) proposes a framework that allows the agent to adjust its actions based on the effects of those actions. Shinn et al. (2023) introduces reinforcement learning to refine the adjustment. However, this method requires immediate rewards, which are not available in our setting. Press et al. (2023) encourages the agent to ask questions that are beneficial to problem-solving. Wang et al. (2023a) allows the agent to devise a plan before trying to solve a problem. However, the questions and steps in such plans are usually simple fact checks or atomic operations such as calculations, which are not essential for open-ended analysis.

**Time-series Forecasting in Social Science.** Predictability in finance has been studied under the efficient market hypothesis (Fama, 1970) which implies that it takes time for information to be reflected in the price, leaving room for predicting the price at some future time. Financial prediction has been studied in various fields, such as machine learning (Krollner et al., 2010; Kelly & Xiu, 2023; Kim, 2003), deep learning (Sezer et al., 2020), LLMs (Lopez-Lira & Tang, 2023), and natural language processing, which is used to capture sentiment levels and other features (Xing et al., 2018; Lukauskas et al., 2022). Unlike pure time series forecasting, SocioDojo prohibits "day trading", forcing the agent to predict long-term trends that rely on information beyond historical time series data towards a stronger form of the efficient market. Time series forecasting has also been studied in economics (Granger & Newbold, 2014), polls (Lewis-Beck, 2005), and search trends (Xu & Berkely, 2014; Choi & Varian, 2012). To make reliable predictions on these topics, it is necessary to access a variety of sources of information, which can be found in SocioDojo.

## 3 SOCIODOJO

In this section, we first formulate the Hyperportfolio task in the Partial-observable Markov Decision Process (POMDP) framework in Section 3.1. Then we introduce the construction of SocioDojo, which serves as the environment for the POMDP in Section 3.2.

### 3.1 HYPERPORTFOLIO TASK

Most existing learning environments are designed for embodied agents that learn to take the right sequence of actions to interact with the environment, which can be smoothly assessed by checking the immediate outcomes of the agent's actions. On the contrary, our aim to comprehend and analyze society is not easily quantifiable. People use exams to measure such capabilities, but creating good exam questions necessitates the involvement of experts and regular updates to keep up with the ever-evolving society. We therefore propose a **hyperportfolio** task, based on POMDP, which (1) evaluates agents in a scalable manner using the ever-updating real-world time series, and (2) provides a reliable indicator of the level of analytical power and understanding of the real society.

### 3.1.1 SOCIETY UNDERSTANDING AS DECISION-MAKING

The society can be modeled as a hidden Markov process $P(\omega_t|s_t)$, where $s_t$ is the current hidden world state, $\omega_t$ is the observation. Understanding society is to estimate the hidden world state denoted as $P(s_t|\omega_t)$. Humans perceive society through multiple sources of information, such as

newspapers, social media, websites, etc. The world state, on the other hand, is portrayed by the time series of different facets of society. For instance, economic and financial time series illustrate production, trade, consumption, activities, etc., search trends uncover the public's interests, and political and opinion poll trackers demonstrate people's views on politics and public issues. These time series were designed to probe the world state as accurately as possible through statistical techniques and social science studies. Therefore, we can regard those high-quality time series as a near approximation of the hidden world state. We use *probe set* $Y_t \in \mathcal{R}^N$ consisting of the *latest* values of a set of time series at time $t$, where $N$ is the number of selected time series, as the approximate current world state. Consequently, estimating the probe set can be used as a substitute task for the estimation of the current hidden-world state, where $P(s_t|\omega_t) \approx P(Y_t|\omega_t)$.

However, it is not possible to accurately estimate the current probe set given the current observations. Probe values are delayed, denoted as $P(y_t|s_{t'})$ where $y_t \in Y_t$ and $t' < t$. Economic statistics and public polls take time, financial data need time to reflect the latest information, and search trends need to wait until a topic is broadcast before it becomes a spike in the time series. The observations are also delayed, denoted $P(\omega_t|s_{t''})$ where $\omega_t \in \Omega_t$ and $t'' < t$. Since news cannot be perfectly real-time due to production and delivery time. Furthermore, a newly published message may not necessarily talk about the latest world. As a result, the probes and observations are mismatched, and we can only approximately estimate the probe value at an undetermined future time point.

Inspired by portfolio optimization, we convert the precise estimation problem to a surrogate decision-making task, the **hyperportfolio**, which largely relies on the estimation power of the world state, i.e. the capacity to understand and analyze society. Modern portfolio theory has been introduced in finance to maximize the expected return of a combination of financial assets expressed as $\max E(R_p) = \sum_{i=1}^n w_i E(R_i)$, where $w_i$ is the proportion of the $i-$th asset in the total investment value, $R_i$ is the return of the asset $i$, its definition can be complex, a simple model is $\frac{y_i^{sell} - y_i^{buy}}{y_i^{buy}}$, where $y_i^{buy}$ is the buying price of an asset, $b$ denotes the buy time, and $y_i^{sell}$ is the selling price, the price is the value in the time series with buying or selling time.

In hyperportfolio, all high-quality time series that accurately probe the world state, not only financial ones, are investable assets. The current price of an asset is determined by the *most recent* value in the corresponding time series. An agent needs to make investment decisions over time and maximize its assets to show its analysis power and understanding level of society. As we assume wise investment decisions that maximize the expected return should rely on a good estimation of the dynamics of the world state. To ensure this condition, we design mechanisms that will be discussed later in Section 3.2.2 based on our robustness analysis in Appendix C, to reduce the chance of unwanted speculation activity and the factor of "luck". This task is easily scalable, as time series, information sources, and knowledge bases are constantly updated in the real world, which also ensures that the agent developed in SocioDojo can be immediately applied in real life.

### 3.1.2 POMDP OF HYPERPORTFOLIO

A hyperportfolio $H = \{(c_i, w_i)\}$, where $c_i \in C$ is the ID of an asset, in SocioDojo, each time series has a unique ID, $C$ is the set of all IDs, and $w_i$ is the ratio of this asset in the portfolio. For convenience, we consider cash $(c_0, w_0)$ an asset in a hyperportfolio whose values do not change over time for simplicity. The hyperportfolio for each agent will be initialized as cash only, which means $H_0 = (c_0, 1)$. We formulate the hyperportfolio task in POMDP

$$(S, A, T, R, \Omega, O, \gamma, Z) \tag{1}$$

where $S$ is the hidden **state space**, $\Omega$ is the **observation space**, $A$ is the **action space** contains the hyperportfolio adjustments $(c_i, \lambda), i > 0$, where $-1 \leq \lambda \leq 1$ is the ratio to the total asset, the sum of cash and the portfolio value. A positive $\lambda$ indicates *"buy"*, which requires $c_i \in C$, and $\lambda_i \leq w_0$ to ensure there is sufficient cash. A negative $\lambda$ represents *"sell"*, which requires $c_i \in C_H$, the holding assets, and $\lambda_i \leq w_i$, where the sell value should not exceed the owned value. $\lambda = 0$ means no action. We do not introduce some financial actions like *leverage* and *short* for simplicity. The **reward** $R = E(R'_h) - E(R_h)$ is the change in the expected returns of the hyperportfolio after taking action. $T$ is the **conditional state transition function**, we assume $T(s^{t+1}|s^t, a^t) = T(s^{t+1}|s^t)$ where actions do not impact the transition of the world state which is driven by the human society itself. $O$ is the **conditional observation function**; similarly, we assume that $O(o^t|s^t)$, which is

driven by the media in the real world, is independent of the action. The **discount factor** $\gamma$ is 0 since we do not care about historical improvements to the hyperportfolio. Additionally, **context** $Z$ is introduced, as the observation may not contain the necessary context for analysis.

## 3.2 SOCIODOJO ENVIRONMENT

SocioDojo uses three components - Information Sources, Time series, and Knowledge base & Tools - based on ∼30 GB of high-quality real-world data that we have collected, to create a simulated environment for POMDP in equation 1. With these building blocks, two key modules, the *World Runner* and the *Broker*, are used to implement the work process of the hyperportfolio POMDP. Details about how we reduce biases and filter sources can be found in the Appendix B.

### 3.2.1 COMPONENTS

**Information Sources.** This component composed the observation space $\Omega$ with high-quality real-world media collected from the Internet and manually filtered to reduce noise and bias between October 1, 2021 (the date of the most recent knowledge of GPT-3.5 and GPT-4) and August 1, 2023. Sources cover various types of information, such as newspapers, social media, public reports and insights, research papers, website articles, etc. The distribution of types and topics for each type can be found in Table 1 and Appendix A.1 respectively, while Appendix A.2 shows the distribution of topics of messages over time. The information sources were stored in a unified structure (`index,metadata,messages`) called *channels* which include the datetime, metadata, and contents of the messages in each channel, respectively. Agents can customize information sources as a **queue** of messages (`datetime,metadata,content`), ascending ordered by time, with channels. We follow the terms of use to curate our data, and for the few copyrighted contents that did not allow distribution, we provide metadata including the content URLs to minimize the effort for researchers to rebuild the environment in their local settings.

| Type | Num | Avg. Tok. | Avg. Per day |
|---|---|---|---|
| Newspapers | 175.2K | 674 | 262 |
| Social media | 259.8K | 50 | 388 |
| Reports | 19.5K | 4926 | 29 |
| Papers | 6.2K | 26939 | 9 |
| Other | 32.7K | 1596 | 49 |

Table 1: Statistics of information sources by type.

| Type | Keys | Tokens | Avg. Tok. |
|---|---|---|---|
| Encyclopedia | 57.4K | 14.8M | 258 |
| Books | 167.5K | 41.0M | 245 |
| Papers | 1.7M | 392.4M | 228 |

Table 2: Statistics of knowledge base by type.

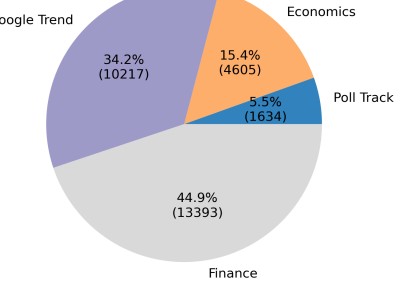

Figure 2: Statistics of time series by topic. Number of time series shown in the bracket.

**Real-world Time Series.** This component provides the probe set $Y$ discussed in Section 3.1.1 to approximate the world state $S$. We collect time series on a variety of topics for a comprehensive probe of the world state, including financial data from Yahoo Finance, economic time series from the St. Louis Federal Reserve Economic Data Database (FRED) with a popularity rating of more than 50%, Google trends of free trending keywords from Exploding Topics, a society trend tracking service, political polls from FiveThirtyEight, a famous political analysis website, and public opinion poll trackers from YouGov, an online survey platform. We remove any time series that contain nonpositive values. We also rescale Google trends with a random number from 0.1 to 10, since Google trends use a relative scale from 0 to 100 which could potentially leak future information to the agent. The distribution of topics is shown in Figure 2 and is detailed in the Appendix A.3.

**Knowledge Base & Tools.** This component complements the agent with the necessary context $Z$ that is not available from the input messages. Open resources on the Internet such as Fincyclopedia, Concise Encyclopedia of Economics, Ballotpedia, textbooks, and journal articles on economics,

finance, business, society, etc. are used to build the knowledge base. The texts are segmented, randomly shuffled, and ready for the retrieval-based knowledge base to avoid the distribution of original copies. The toolbox includes search engines and knowledge graph interfaces, which filter search results by current time during the hyperportfolio task to avoid leaking future information. The statistics of the knowledge base can be found in Table 2 and further detailed in Appendix A.4.

### 3.2.2 WORK PROCESS

**World Runner.** World Runner consistently pops the messages from the message queue discussed above. When a message pops out, the world runner sets the world time to the published time of the message and sends it to the agent. The agent runs its internal loop, which involves access to the knowledge base, historical time series, and tools, and interacts with the *Broker* module to send its trading instructions. The world runner pops all messages up to the end time, i.e. Aug. 1, 2023, from the queue and then calculates the hyperportfolio returns at the end time as the result.

**Broker.** The broker manages the "account" for the agent, executes the trading instructions, and returns the results, including potential errors, to the agent. We introduce 4 mechanisms to alleviate undesired returns: *Forbid day trading*, *Commission fee*, *Overnight fee*, and *Return bound*. The detailed definition of these mechanisms, as well as our study of the volatilities of time series and random walk tests to find the optimal overnight rate and bound, can be found in the Appendix C.

## 4 AGENT ARCHITECTURE AND HYPOTHESIS & PROOF PROMPTING

We present our Analyst-Assistant-Actuator (AAA) agent architecture to tackle the hyperportfolio task in Section 4.1 and an overview can be found in Appendix E. Subsequently, we discuss the Hypothesis & Proof prompting technique for generating high-quality analysis in Section 4.2.

### 4.1 ANALYST-ASSISTANT-ACTUATOR ARCHITECTURE

We break down the analysis and decision-making process into three distinct roles with varying duties and access to the three components of SocioDojo: analyst, assistant, and actuator. To (1) save tokens for each role, (2) have each role concentrate on a particular task with a precise context, which could enhance the precision of the subtasks, and (3) provide better flexibility, we can use different foundation models for different roles to save cost, we do an ablation study for this in Section 5.3.

**Analyst.** The analyst is the entry for messages. First, it reads the metadata, which may include the title, description, and category, to determine whether it is necessary to read the full content or not to save costs. If it is decided to read the full content, the analysis loop, a multi-round dialogue between the analyst and the assistant, will begin. The analyst can use the `ask` function to request the assistant to seek information to complete the analysis and the `done` function to finish the analysis. The system will then prompt the analyst to provide a final analysis report. Finally, the analyst can decide whether to send the report to the actuator or not, depending on whether there is any immediate useful information for the actuator to adjust the hyperportfolio. The user can specify a maximum number of steps beyond which the agent must provide its final report to balance cost and performance. In our experiment, we set this number to 4.

**Assistant.** The assistant has access to the knowledge base, tools, and historical time series. It answers queries ranging from simple data requests to more complex questions such as "Will some event impact the economy?". The query is handled iteratively through a multi-round dialog response loop with a max step of 3 in our experiment. It can call one of the query interfaces to find resources at each step. The knowledge base and the time series base are implemented as two vector databases by ChromaDB with Instructor-XL (Su et al., 2023) as the embedding model. The natural language query is embedded by the model, then its inner product distance is computed to the embeddings of snippets stored in the database, and the top candidates are returned. For the knowledge base, the snippets are segmented texts discussed in Section 3.2.1; for the time series, the snippet is the code of a time series and its description, allowing the agent to find the code of a desired time series. When the assistant gets sufficient information, it can call the `done` function to return the final answer.

**Actuator.** The actuator interacts with the broker. It initiates a multi-round dialog action loop with a max step of 5 in our experiment when an analysis report is received. It can query the assistant or send trading instructions, which are written in a multiline script. Each line of the script follows the format `BUY/SELL [CODE] [VALUE]`, and the `done` function is called to complete the loop.

## 4.2 HYPOTHESIS & PROOF PROMPTING

Existing prompting techniques optimize reasoning in scenarios where the query can be broken down into clear procedures. For example, Self-Ask (Press et al., 2023) encourages the agent to ask questions to collect information during reasoning, while Plan & Solve (Wang et al., 2023a) guides the agent to devise a plan to address the problem. Such an iterative approach encourages a more thorough reasoning process. Nevertheless, the analysis process cannot be partitioned into distinct steps, and it is not clear what questions can enhance the depth of analysis which can extrapolate more useful indicators that can direct investment decisions in the context of hyperportfolio; moreover, it is hard to devise samples that prompt the agent in a few-shot way like most existing methods.

Inspired by the scientific research process, we propose the zero-shot **Hypothesis & Proof** (H&P) prompting. This meta-thinking framework encourages the agent to come up with hypotheses beyond the domain of the input and to search for evidence to prove or disprove them. The prompting is provided as additional context for the analyst, who will make hypotheses, such as "cold weather will reduce people's willingness to shop", and ask the assistant to find evidence to support or refute it. The analyst will then decide whether to accept or reject the hypothesis based on the evidence returned. If there is no sufficient support, the analyst can choose to keep it as an intuition. In the final report, the analyst will clearly list the hypotheses, proofs, intuitions, and references to the evidence. Detailed promptings can be found in the Appendix F.

We provide an example of a full H&P analysis process and compare it with the P&S prompting in Appendix G. The analyst can iteratively propose more in-depth hypotheses that gradually uncover deeper insights from the input step by step. This allows the analyst to be more sensitive and notice subtle trends that may be overlooked by other analysts who only ask fact-search questions. We evaluate them and discuss the results in Section 5.2.

## 5 EXPERIMENT

In Section 5.1, we present our experimental setup. We then evaluate our proposed H&P prompting and compare it with other state-of-the-art prompting techniques in Section 5.2. Finally, we discuss the results of the ablation studies in Section 5.3.

## 5.1 EXPERIMENT SETTING

We designed two experiment settings. *Strandard* allows the agent to freely configure its hyperportfolio and *Tracking*, which limits the assets of the hyperportfolio to a list of 100 selected time series, including the Dow Jones 30 and other popular financial assets, 20 most popular time series in FRED, national polls, and a few Google Trend keywords, to evaluate the ability of agents to comprehensively analyze the major trends of society. The list is provided with the dataset.

We compare H&P with the following state-of-the-art methods. All methods use AAA architecture, one news channel as the information source, `GPT-3.5-Turbo` series as foundation models with a low `temprature=0.2` for a more deterministic experiment result.

**Plan & Solve** (P&S) (Wang et al., 2023a) guides the agent to design a plan first and then execute it to solve a problem. P&S is zero-shot, so we consider it as a major baseline and utilize it in the analyst by replacing "answering the question" with "analyzing the article" in the prompt.

**Self-Ask** (Press et al., 2023) is a few-shot prompt that leads the agent to ask questions to search during question-answering. We use it with assistant as few-shot samples for the analyst are unclear.

**ReAct** (Yao et al., 2023c) allows the agent to refine its actions by observing the results of previous activities, such as searching or using tools. ReAct is also few-shot, thus we employ it in the assistant.

| | | H&P (Ours) | | P&S[1] | Self-Ask[2] | ReAct[3] | AutoGPT[4] | Vanilla | |
| | | full | w/o AAA | | | | | w/ Ask | w/o Ask |
|---|---|---|---|---|---|---|---|---|---|
| S. | Fin | **15.3** | 13.0 | 12.4 | 11.8 | 10.9 | 9.8 | 10.4 | 6.1 |
| | Econ | **5.5** | 4.5 | 1.0 | 3.6 | 2.2 | -6.3 | 1.8 | 0.2 |
| | Web | **15.0** | 13.8 | 11.1 | 9.4 | 9.5 | 8.3 | 10.2 | -4.5 |
| | Poll | 3.6 | 2.2 | **4.1** | -2.9 | 1.7 | 3.1 | -1.2 | 2.5 |
| | All | **14.3** | 11.9 | 10.8 | 9.9 | 9.4 | 8.9 | 9.8 | 3.1 |
| | *S* | *3.21* | *2.91* | *2.90* | *2.60* | *2.15* | *2.77* | *2.16* | *1.20* |
| T. | Fin | **15.6** | 13.9 | 12.0 | 11.6 | 10.3 | 8.6 | 10.7 | 8.0 |
| | Econ | **2.8** | -4.2 | -3.2 | -1.1 | -1.7 | 1.0 | N/A | 0.2 |
| | Web | 13.7 | **14.9** | 8.7 | 4.3 | 4.9 | 13.3 | 3.8 | 7.3 |
| | Poll | **3.1** | 2.7 | N/A | 1.4 | -1.1 | 2.3 | 0.1 | -6.7 |
| | All | **15.0** | 13.1 | 11.5 | 10.6 | 8.8 | 10.3 | 10.0 | 6.6 |
| | *S* | *3.07* | *2.42* | *2.68* | *2.15* | *2.43* | *2.45* | *2.35* | *1.67* |

Table 3: Percentage returns in hyperportfolio task of agents including ours, 1. Wang et al. (2023a), 2. Press et al. (2023), 3. Yao et al. (2023c), 4. Richards (2023), and a vanilla analyst without additional prompting. "S." and "T." represent "Standard" and "Tracking" settings respectively. "AAA" means Analyst-Assistant-Actuator architecture. "Ask" means whether the agent can access the Internet and knowledge base. "All" means the overall return, and "Fin", "Econ", "Web", "Poll" denotes the returns of the assets in those topics. "N/A" means the agent does not invest assets in that topic. "*S*" is percentage Sharpe ratio, as a reference, S&P 500 in the same period is 1.53, Apple stock is 4.73.

**AutoGPT** (Richards, 2023) is the state-of-the-art LLM agent framework that encapsulates tools and prompts that allow automatic planning and completion of a given task, which is analysis in our case.

We also tested our method without the AAA architecture, in which a single agent plays all roles, a vanilla analyst without any extra prompting and one without access to the knowledge base & tools.

## 5.2 EVALUATING AGENTS

Table 3 displays the results. Our H&P prompting had the highest returns with 14.3% for the standard setting and 15.0% for the tracking setting, which is a 32.4% and 30.4% improvement over the P&S, the second-best performing method. H&P also outperforms most of the topic-wise returns. We also provide the Sharpe ratio to measure risk-adjusted returns by $S = \frac{(\bar{r_p} - \bar{r_f})}{\sigma_p}$, where $\bar{r_p}$ and $\sigma_p$ are the mean and standard deviation of the daily percentage change in the hyperportfolio value, and $\bar{r_f}$ is the risk-free return, which is assumed to be 0 in our setting (see Appendix C). It is important to note that SocioDojo uses publicly available daily close data for financial assets, which cannot be directly interpreted as or compared to real financial returns. We discuss the potential impact of this and the returns adjusted by the commercial minute-level financial data from AlphaVantage in Appendix D.

Both H&P and P&S benefited from augmenting the analyst, which can be attributed to their iterative approach. H&P further encourages the agent to explore more deeply by forming out-of-domain hypotheses. For example, H&P learns to compare similar situations in history to discover potential indicators that are not directly available from the input, as shown in Appendix G. In contrast, P&S does not ask questions outside of the input. Specifically, a H&P analyst is likely to ask ques-

| | H&P | P&S | Other |
|---|---|---|---|
| Mean | 1.91 | 0.72 | 0.52 |
| Std. | 0.98 | 0.88 | 0.85 |

Table 4: Number of `ask` calls made by agents in our experiments.

tions such as *"what is the potential impact of X on Y?"* or *"what is the significance of X in Y?"*, while baselines tend to ask questions such as *"what is X?"*. The statistics in Table 4 demonstrate that our H&P agent made a significantly higher number of valid `ask` calls, indicating that it is able to form more in-depth hypotheses. In contrast, the P&S agent only had a slight increase in `ask` calls, likely due to its inability to construct useful questions. Other baselines augment the assistant. However, the analyst's lack of in-depth questioning and early termination of the analysis prevented the

improvement from being demonstrated. Furthermore, the vanilla agent without access to the knowledge base and tools performs poorly due to the lack of the latest background of the current world, for example, the world before and after COVID is largely different. Thus, we can conclude that the analyst plays a key role in making high-quality plans and utilizing the assistant more effectively.

## 5.3 ABLATION STUDY

| Type | Fin | Econ | Web | Poll | All |
|------|-----|------|-----|------|-----|
| News. | 15.3 | 5.5 | 15.0 | 3.6 | 14.3 |
| +Social. | 14.3 | 5.3 | 15.3 | 2.3 | 14.1 |
| +Report | **15.9** | 6.3 | 14.2 | 3.9 | **14.7** |
| +Papers | 15.3 | **6.7** | 15.0 | 3.4 | 14.6 |
| +Other | 15.0 | 5.3 | 14.9 | **4.9** | 14.4 |
| Mixed | 15.5 | 5.8 | **15.4** | 4.0 | **14.7** |

Table 5: Ablation study of types of sources.

| Model | Fin | Econ | Web | Poll | All |
|-------|-----|------|-----|------|-----|
| GPT-3.5 | 15.3 | 5.5 | 15.0 | 3.6 | 14.3 |
| GPT-4 | **17.2** | **7.3** | **16.7** | 4.2 | **16.8** |
| Assis. | 15.9 | 5.8 | 16.4 | 4.3 | 15.2 |
| Actua. | 15.8 | 6.0 | 15.3 | 3.1 | 14.9 |
| Analy. | 16.9 | 6.8 | 16.2 | **4.4** | 15.9 |

Table 6: Ablation study on foundation models by replacing "GPT-4" for all roles or for each of "Assistant", "Actuator", "Analyst".

**Information sources.** We add additional sources of different types to the basic setting above. The topic distribution of each type can be found in the Appendix A.1. Table 5.3 shows the results, which indicate that the inclusion of more information sources does not necessarily lead to better performance. This could be due to the fact that different sources require different analysis techniques; for instance, reading papers is different from reading news, and some messages, such as papers, were truncated to fit the token length limit. Furthermore, information sources may also introduce noise, such as fake news or unverified opinions in social media, even though we have filtered out relatively reliable publishers. Therefore, we conclude that high-quality information sources are essential, and domain-specific analysis techniques are necessary.

**Foundation models.** We substitute the foundation model with `GPT-4` for all roles or for each role. The results are presented in Table 5.3. We can observe a marked improvement over `GPT-4`, which can be attributed to its more comprehensive knowledge of social science, as evaluated by multiple human tests (OpenAI, 2023). We also note that the most significant improvement is seen in the analyst, which is the main bottleneck. Enhancing the actuator also yields an improvement, though not as prominent, indicating that the bottleneck does not come from the actuator. In contrast to baselines, improving the assistant gives a boost, which can be explained by the fact that the questions asked by a H&P analyst are more complex and require more capability. Therefore, we conclude that the analyst is critical for the hyperportfolio task and also has an effect on the gains from the assistant, while a well-executed actuator is also beneficial but not the highest priority.

## 6 CONCLUSION

In this paper, we introduce SocioDojo, a lifelong learning platform for developing ready-to-deploy agents that are capable of analyzing and making decisions about the dynamics of human society. SocioDojo is composed of real-world news, articles, papers, etc. over time from diverse information sources, time series from economics, finance, search trends, and polls that support the hyperportfolio, a decision-making task that can evaluate agent analysis and decision-making power in a reliable and scalable way, and a knowledge base of books, journals, encyclopedias along with the search interfaces. Our Analyst-Assistant-Actuator architecture and Hypothesis & Proof prompting achieve improvements of 32.4% and 30.4% in two experiment settings compared to the state-of-the-art methods, which is a promising start for the development of lifelong analytical agents for human society that aim to achieve human-centered AI.

## 7 REPRODUCIBILITY STATEMENT

We ensure the reproducibility and completeness of this paper by introducing the hyperparameters and experiment settings in the main paper. To make the results more deterministic and reproducible,

we use a low temperature for LLMs. We also design mechanisms to make the results stable and provide details including two key hyperparameters, overnight rate, and return bound in the Appendix C. Furthermore, we provide all the promptings for our architecture and details about the analysis process in Appendix F. To enable readers to reproduce our work, we make our code and data publicly available.

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

# A SOCIODOJO STATISTICS

## A.1 DISTRIBUTION OF TOPICS FOR EACH TYPE OF INFORMATION SOURCE

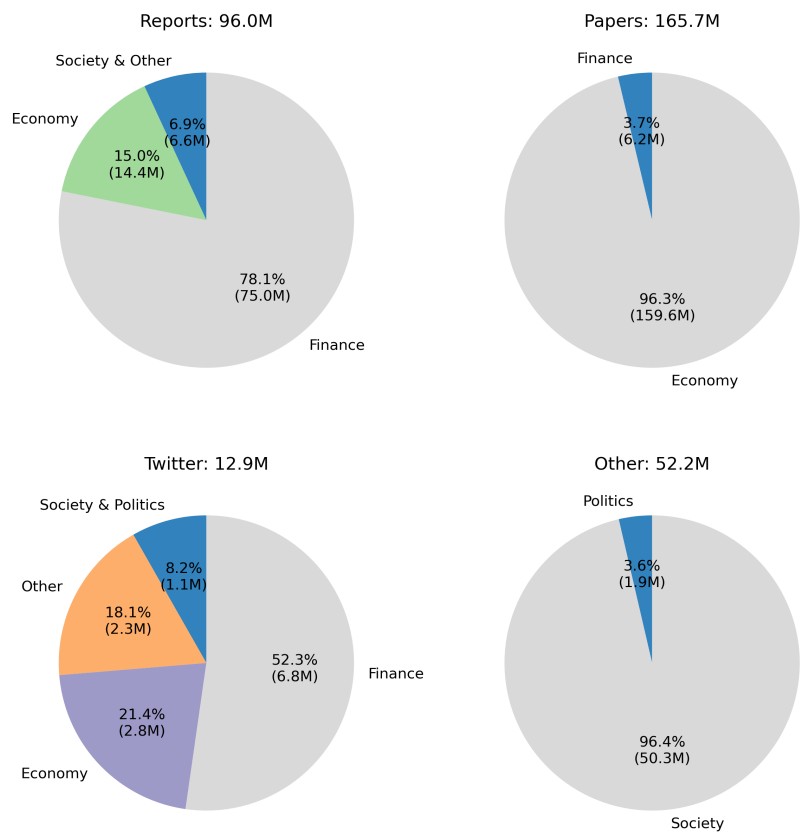

Figure 3: Distribution of topics for each type of information source by number of tokens.

## A.2 DISTRIBUTION OF TOPICS FOR INFORMATION SOURCES OVER TIME

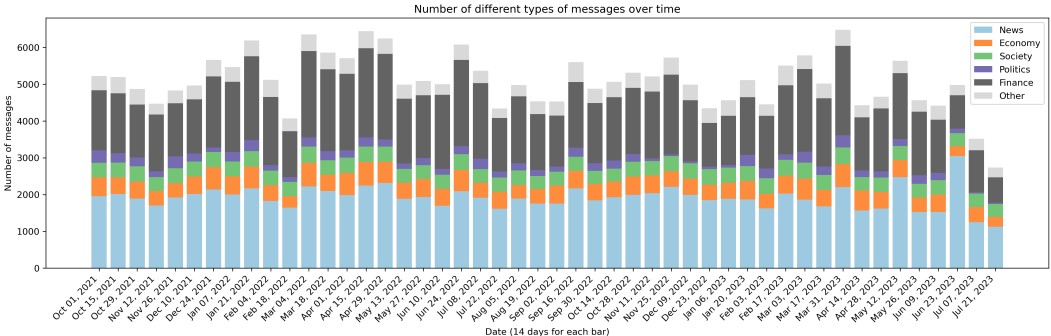

Figure 4: The x-axis indicates the date, and each bar represents the distribution of the number of messages for each topic over a 14-day period beginning with that data. The graph displays how the topics of messages have been distributed over time in all information sources.

## A.3 Distribution topics of time series for each domain

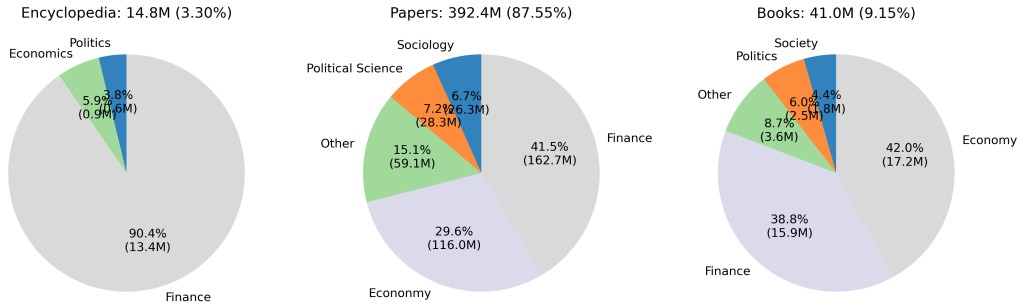

Figure 5: Distribution of topics in different domains of time series by number of time series.

## A.4 Distribution of topics for knowledge base sources

Figure 6: Distribution of topics for each knowledge base source by number of tokens.

## B Methods in Creating SocioDojo

News media have political biases, so to reduce the influence of these biases on our information sources, we refer to the Media Bias Ratings and Charts from Allsides, which provides a rating from -6 to 6. We only choose media with ratings between -3 and 3, and as close to neutral as possible. We also balance the ratings by letting the summation of the ratings of all selected media be as close to 0. Additionally, we refer to the media rank from Stony Brook University to pick highly ranked

media. We use Google Scholar venue rankings to screen for reliable sources of academic papers. We filter social media accounts by the number of followers and then manually remove accounts that show clear biases. We manually search for reputable institutions in different domains to obtain their publicly available reports. We also select textbooks from the Internet that have good ratings. To prevent information leakage, the materials in the knowledge base are published before the start date of the environment 2021-10-01, and we assume that the knowledge base information is timeless and valid throughout the experiment period as we manually pick only the highest-rated sources from the Internet. Finally, we also consulted experts and librarians in economics and other domains to verify the reliability and diversity of our applied sources.

## C    ROBUSTNESS ANALYSIS OF SOCIODOJO

Robustness is used to describe the ability of a system to withstand disturbances that could potentially affect its functioning. In the context of SocioDojo, which is responsible for assessing the societal analytical and decision-making capabilities of agents, robustness can be determined by whether an agent can achieve a high return without relying on high-quality analysis and sound decisions.

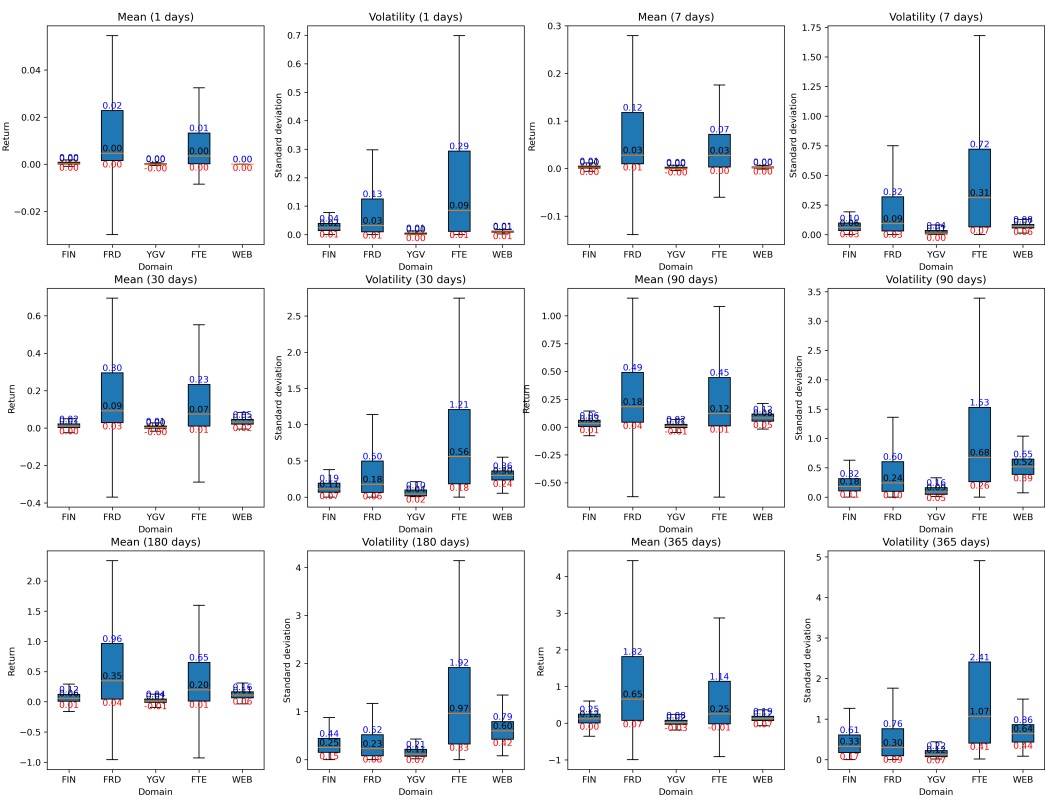

Figure 7: The volatility and mean changes under different sampling periods for time series in different sources. 0.1 on the y-axis means 10% return.

We introduce four mechanisms to alleviate undesired returns: (1) *Forbid day trading*, that an asset cannot be sold within 5 days of purchase to avoid profiting from short-term patterns. Moreover, it balances the agent's information advantage as we use close prices in finance which do not reflect the latest news in real-time. (2) *Commission fee*, charged for each transaction to avoid frequent trading and encourage considered decisions. We use 1% of the trading value in our experiments. (3) *Overnight fee*, calculated as $rate \times size \times y^t/360$ where $size = v_{base}/y^{buy}$ and $v_{base}$ are the cash used to buy the asset, charged daily for some assets. Since some assets are almost ever-growing, such as GDP, which leads to risk-free returns, we wish all returns to come with risk, thus requiring good analysis to achieve a better risk-return trade-off. We use 0.1 for economics and 0.05 for search

trends. (4) *Return bound*, limited the returns of an asset by $-bound \times v_{base} \leq R_i \leq bound \times v_{base}$, thus avoiding extreme value and alleviating speculation from high volatility. We use a bound of 5 in our setting.

We begin by studying the volatility of the time series from various sources, as seen in Figure 7. FIN stands for finance, FRD is the economic time series from the FRED database, YGV is the poll trackers from YouGov, and FTE is the political poll data from FiveThirtyEight. The "Poll" type in the SocioDojo time series is a combination of the YGV and FTE time series. WEB is Google Trends.

Volatility is a measure of the standard deviation of changes in a time series with a given sampling period, such as the standard deviation of the weekly change. It is calculated as $\sigma_P = \sqrt{\frac{1}{N_P} \sum_{i=d_0}^{d_n - P} (v_i^P - \bar{v_P})^2}$, where $d_0$ is the first day and $d_n$ is the last day of the time series, $P$ is the sampling period, $v_i^P = y_{i+P} - y_i$ is the change in value during a period of length $P$ beginning on the day $i$, $N_P$ is the number of spans under the period of $P$ in this time series and $\bar{v_P} = \sum_{i=d_0}^{d_n - P} \frac{v_i^P}{N_P}$ is the mean change under the sampling period of $P$. High volatility usually indicates a high risk, while a high mean usually implies a high potential return.

FRD shows high means for all its quartiles as some FRD time series, such as GDP and population, are almost ever-growing, which presents both a risk to robustness that an agent may randomly invest in an economic time series and still get a good return without conducting a thorough analysis. This has motivated us to introduce an overnight rate to balance the risk-reward trade-off and reduce speculation. We also introduce an overnight rate for the WEB since its high volatility leaves a potential risk of speculation. However, the excessive volatility of FTE makes it too high risk, and YGV has low volatility and return. Thus, we impose *negative* overnight rates of 0.05 on them to rebalance risk and returns. We also do not apply overnight rates on FIN to align with the real financial market.

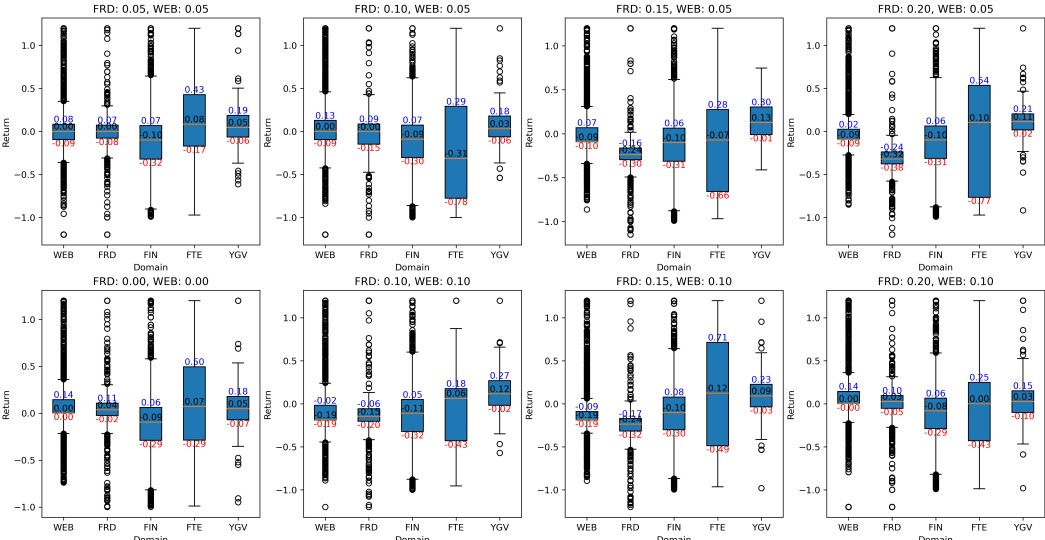

Figure 8: Distribution of returns of random walk agent under different overnight rates for FRD and WEB, the returns are truncated around -/+ 120%, 0.1 in y-axis means 10% return. 100 runs for each setting.

The level of robustness for SocioDojo can be depicted by $E(R_u)$, the expected return of unexpected behaviors. We use the random walk agent whose expected return should ideally be 0 to explore the robustness of SocioDojo and find a suitable overnight rate. The agent sampled evenly from all assets to buy with a value of 5000 to 50,000 and from owned assets to sell with a value of 0 to the owned value in each step and had a probability of 0.5 to make no move. The random agent accepted the same information source as the one used in the agent evaluation experiment in Section 5.2, although the message was not actually read, to maintain the same action frequency. Different overnight rates

of FRD and WEB were tested, and each setting was run 100 times. The distribution of the returns is visualized in Figure 8.

The WEB, FRD, and FIN assets display many outliers, so a boundary would significantly reduce the likelihood that the agent will receive a high reward by chance. `FRD: 0.05, WEB: 0.05,` and `FRD: 0.10, WEB: 0.05` achieve an ideal mean return of 0. We can also observe that the returns of FTE and YGV have a relatively high variation; however, due to the negative overnight rate, they also have a relatively higher mean return. The FIN has a relatively high risk and a low mean return; this is because the FIN includes all kinds of financial assets, including high-risk assets such as cryptocurrency and pink stocks. Nevertheless, the nonrandom agents tend to invest in more mainstream companies due to their greater presence in the information sources in practice; thus the return has a large difference compared to a random-walk agent that evenly samples from the assets.

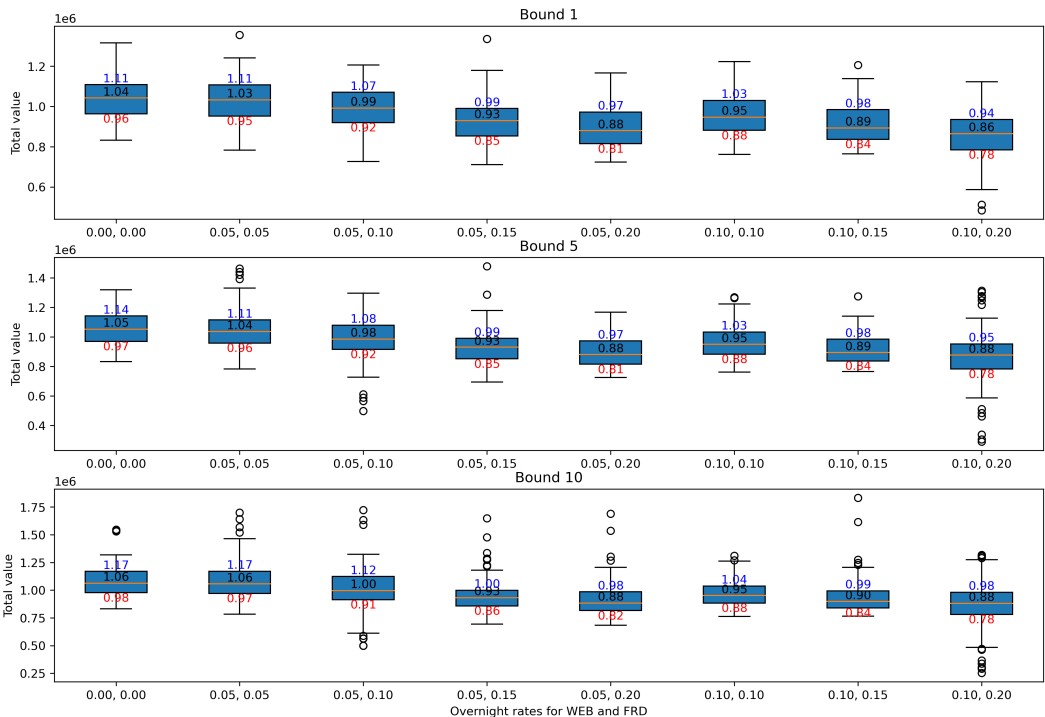

Figure 9: Distribution of final total value of random walk agent under different bound and overnight rates for WEB and FRD. The x-axis shows the overnight rates for `WEB, FRD`. 100 runs for each setting. Each run starts with `1e6` cash.

We further visualize the total return under different overnight rates and bounds in Figure 9. We can see that a lower bound results in fewer outliers. We chose a bound of 5, as it is possible to achieve a return of 5 times for a single asset without leverage in around 2 years, which is the time span of SocioDojo. A bound of 10 is less robust and 1 is too strict. The overnight setting `FRD: 0.10, WEB: 0.05` shows a mean final value close to `1e6`, which corresponds to an ideal zero return. Therefore, we selected overnight rates of `FRD: 0.10, WEB: 0.05`, and a bound of 5 as our experimental setting. Although our designs already provide good robustness, a better overnight rate model based on advanced pricing theories can further enhance the robustness, and factors such as dividends should also be taken into account. We leave them for future work.

# D ANALYSIS OF FINANCIAL TIME SERIES RESULTS

SocioDojo utilizes publicly accessible daily close price data for financial assets, as minute-level historical data usually necessitate commercial licenses to access, and we assess the impact in Section D.1 by adjusting the financial returns with minute-level data. We then employ the adjusted data to

delve into the returns obtained by the financial assets to gain more insight into the performance of the agents in Section D.2.

## D.1 REAL-TIME DATA ADJUSTED RETURNS

| | | H&P | | P&S[1] | Self-Ask[2] | ReAct[3] | AutoGPT[4] | Vanilla | |
| --- | --- | --- | --- | --- | --- | --- | --- | --- | --- |
| | | full | w/o AAA | | | | | w/ Ask | w/o Ask |
| S. | UA | **15.3** | 13.0 | 12.4 | 11.8 | 10.9 | 9.8 | 10.4 | 6.1 |
| | A | **11.0** | 10.1 | 9.7 | 8.0 | 7.9 | 8.4 | 8.0 | 3.3 |
| | S | ***2.44*** | *1.95* | *2.28* | *2.18* | *1.91* | *1.62* | *1.72* | *1.09* |
| T. | UA | **15.6** | 13.9 | 12.0 | 11.6 | 10.3 | 8.6 | 10.7 | 8.0 |
| | A | **11.2** | 9.9 | 9.0 | 9.0 | 8.2 | 5.3 | 7.5 | 6.6 |
| | S | ***2.36*** | *2.05* | *2.32* | *2.19* | *1.71* | *1.44* | *1.80* | *1.25* |

Table 7: Percentage returns in financial time series before and after adjusted by minute-level data of agents including ours, 1. Wang et al. (2023a), 2. Press et al. (2023), 3. Yao et al. (2023c), 4. Richards (2023), and a vanilla analyst without additional prompting. "S." and "T." represent "Standard" and "Tracking" settings respectively. "AAA" means Analyst-Assistant-Actuator architecture. "Ask" means whether the agent can access the Internet and knowledge base. 'UA' means not adjusted, 'A' means adjusted. "$S$" means the percentage Sharpe ratio of adjusted returns, as a reference, S&P 500 in the same period is 1.53, Apple stock is 4.73.

In order to make SocioDojo available to the public, we have chosen to use the daily close price data for financial assets. This has the risk of giving agents an undesired information advantage, as they will be trading with prices that are not up-to-date and do not reflect the latest news. It can be understood that the agent is always receiving a discounted price. Real-time, minute-level historical financial data is not publicly available and requires a commercial license to access. To evaluate the impact of using close price, we adjust the returns using minute-level data obtained from AlphaVantage, a financial modeling platform that provides intra-day data including also before and after market prices, adjusted for historical splits and dividend events. We change the trading price from the latest daily close price to the latest minute price for each transaction in financial assets, and then recompute the returns for each agent.

The adjusted results are presented in Table 7. After the adjustment, the performance of the agents generally decreased without the information advantage, yet the comparative advantages among them remained. This implies that the daily close price data may not be ideal for developing financial models, but it is acceptable for our purpose of evaluating the agents' understanding and decision-making in the real world. This is further supported by Figure 10, which demonstrates that overall trends and dynamics of the financial asset value are preserved in the unadjusted return of all assets. With the rule of forbidding day trading, the agents cannot solely make profits through short-term arbitrage, but rather through improvements in their analysis ability. This also suggests that the SocioDojo returns only reflect the agents' improvement in understanding and decision-making in the real world, and should not be interpreted as real financial returns.

## D.2 ANALYSIS OF RETURNS

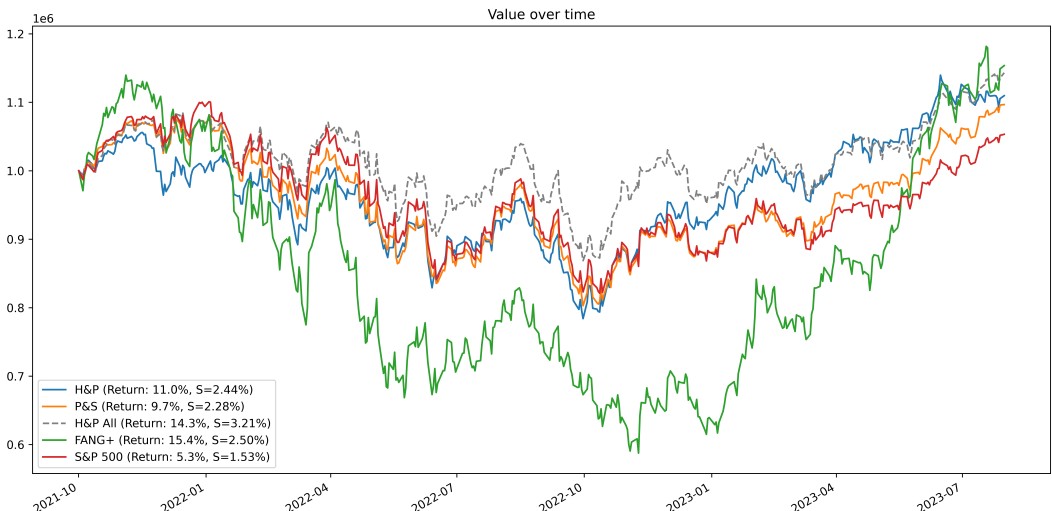

Figure 10: The value of hyperportfolio overtime in the "Standard" setting for our method (grey dashed line) and the value of financial assets after adjustment by minute-level data for our method, Plan & Solve agent, S&P 500, and FANG+ index trackers. We normalize the initial value of all assets as 1 million for a clear comparison.

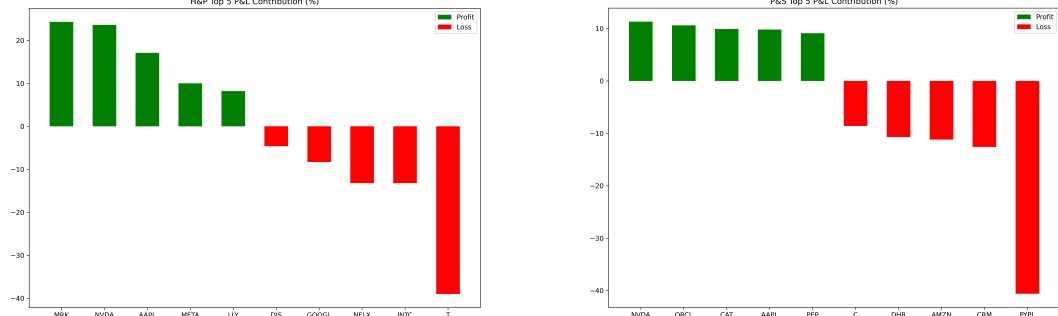

Figure 11: The top 5 gainer and loser stocks that made the most profits and losses respectively in the invested financial assets for our method (left) and Plan & Solve (right). The y-axis denotes the percentage contributions to profit and losses for stocks marked in the x-axis respectively.

We visualize the returns of financial assets over time in the "Standard" setting after adjustment for our method, Plan & Solve, and the baselines including S&P 500 and FANG+ (the index that tracks the top 10 largest technology companies that are also the main sources of returns for agents) in Figure 10. Compared to our method, which performs a deeper analysis, Plan & Solve behaves in a more passive style, highly correlated with the market, as reflected in S&P 500. It follows the major trends obtained from the information sources to make decisions using little personal opinions normally generated through an in-depth analysis. On the other hand, H&P agent is more aggressive, which may be analogous to actively managed funds. This is further evidenced by the top 5 gainers and losers for our method and P&S shown in Figure 11. The distribution of the top 5 gainers in H&P is less even than P&S, indicating a more aggressive and less distributed investment.

# E   OVERVIEW OF ANALYST-ASSISTANT-ACTUATOR ARCHITECTURE

Figure 12: The overview of how the analyst, assistant, and actuator collaborate to analyze and make decisions for one input from SocioDojo.

# F   PROMPTING

## F.1   CONTEXT OF HYPERPORTFOLIO

The context of the hyperportfolio task is shown below. The context will be input as the first system message to the Analyst and Actuator before their role prompts. The Assistant uses a simplified context in its role context that does not contain the detailed rules of the game.

---

**You are a helpful assistant working in a team playing a "hyperportfolio" game that aims to analyze and predict potential movement in real-life social, political, economic, or financial trends from real-time news and other information sources. The rule of the game is as follows:**

1. Every team has an initial account of 1 million cash, your team can use this cash to buy or sell assets to build a hyperportfolio; the target of the game is to maximize the asset, which is the summation of the remaining cash and the value of an asset you own.

2. A hyperportfolio is composed of a set of assets that corresponds to time series in different domains from real-life covers financial market close prices, economic time series, Google search trends, and political and public opinion poll trackers.

3. The game begins on 2021-10-01 and ends on 2023-08-01, after beginning, the time will move forward, and you will consistently receive real-life news about what is happening in the world, newly released opinions from the internet or social network, or reports from research institutes, financial institutions, and so on.

4. Your team may choose to "buy" or "sell" an asset during the game. Each asset corresponds to a time series; the buy price will be the latest value of the time series at the current time.

5. You need to pay a commission when you buy or sell an asset, the amount is about 1% of the buy or sell value.

6. The value of an asset you own will update over time, calculated as (current price/buy price)*(investment amount).

7. For example, you may read news about the Apple company performing well for this season, Based on your analysis, you may think it is a good indicator that Apple stock price will increase and decide to invest 10,000 cash on the Apple stock time series.

---

8. Each time series is marked with an ICode. The ICode has the format "[DOMAIN]:[CODE]". For example, the Apple company stock price is "FIN:AAPL", FIN is the domain, and AAPL is the code. There are five domains, "FIN", "WEB", "FTE", "FRD", "YGV", interpretations for them:

a) FIN: Close price time series of a financial instrument including stocks, ETFs, index funds, REITs, futures, currencies, indices, or cryptocurrencies

b) WEB: The Google trend time series of a keyword, such as "Apple", "iPhone", "Bitcoin", etc.

c) FTE: Political poll tracking time series, such as the president's approval rating, the generic ballot, etc.

d) FRD: Economic time series, such as GDP, PPI, unemployment rate, etc.

e) YGV: Public opinion poll tracking time series, such as support for universal health care, how sustainability policies impact Americans' purchase behavior, etc.

9. You may receive or pay overnight interest or fees if you hold an asset overnight, computed as rate*size*current_price/360, size=amount/buy_price. The rate varies for different assets.

## F.2 ANALYST

Context of the analyst. It will be input to the analyst agent as the second system message after the hyperportfolio context.

**Your role in the group is as an analyst who watches and analyzes the latest information, such as news, articles, reports, etc.**

1. You will work with an actuator who is responsible for managing the hyperportfolio for the team and making buy or sell decisions.

2. Your task is to provide a high-quality analysis for the actuator so that the actuator can make good decisions that optimize the hyperportfolio.

3. You need to find any indicator of potential movement in social, political, economic, or financial trends from the given news.

4. You can give general suggestions, like "Apple stock price will go up", "It is time to sell Apple stock".

5. You can also give more precise buy or sell suggestions if you have confidence that "I think we should spend 10,000 on buying the GDP time series".

6. If you cannot see any opportunity, you should also indicate that "I cannot see any indicators", or "I think we should wait for now".

Hypothesis & Proof prompting. It will be inputted to the analyst agent as the third system message after the analyst context.

**In order to make a convincing analysis, you must clearly state your hypothesis and provide evidence to support your hypothesis:**

1. If your hypothesis cannot be supported by the facts, knowledge, and information you have, you should seek the help of another assistant agent on your team by calling the `ask` function.

2. You should never make up facts, knowledge, or information, if you do not know, you should call the `ask` function to seek help from the assistant agent.

3. The assistant agent will find evidence to support your hypothesis if you call the `ask` function and send the evidence to you. It can also search for general information, facts, and knowledge, as well as information on an ICode.

4. The hypothesis can also be rejected by the evidence; in that case, you should give up the hypothesis and construct a new one.

5. Your hypothesis may lack support for both acceptance and rejection; in that case, you can choose to insist on it as an intuition, but you should explain why you insist on it, or you can choose to give up it.

6. When you are done, you should call the `done` function to finish the analysis.

7. You must call one function `done` or `ask` in each step of the analysis.

**Here are some tips for you to make a hypothesis:**

1. A hypothesis is like a guess, assumption, or intuition, it is a statement that you think is true, but you do not have enough evidence to support it yet.

2. For example, "interest increase in technology will cause the technology stock to increase", "sale of lipstick indicates a potential drop of the economy", "the president's approval rate will increase if the economy is good", etc.

3. The hypothesis can be microscopic or macroscopic, it can be about a specific asset or a general trend.

Now, progressively analyze the given news, article, report, etc. in a multi-round dialog between you and the `ask` function. When you think that you get enough information to give the final analysis report, call the `done` function to end the dialog with the `ask` function, and aggregate the progress as a final analysis report when the system asks you to give your final analysis report. You must call one of the `done` or `ask` functions in your response.

## F.3   ASSISTANT

Context of the assistant. It will be inputted to the assistant agent as the first system message without hyperportfolio context.

You are a helpful assistant working in a team playing a game of "hyperportfolio" that aims to analyze and predict the potential movement in real-life social, political, economic, or financial trends from real-time news and other information sources.

**Your role in the team is the assistant who helps the analyst find evidence to support or reject the hypothesis or search for the facts, knowledge, information, or data demanded by the analyst:**

1. You can access the search engine or database.

2. You need to understand the demand from the analyst and generate the query for the search engine or database that can best meet the analyst's demand.

3. If the request is a hypothesis and you do not get ideal evidence from your search, you may need to reason and decompose it into subhypotheses, and then search for the subhypotheses to find evidence to support the original hypothesis.

4. When you are finished, call the `done` function to complete the analysis.

5. You can give your opinions, but the main role is to give evidence from your search results.

**Here are the tips for using the search engine or database:**

1. You can use Google search by calling the `googlesearch` function to find general information or facts.

2. You can use Google Knowledge Graph Search by calling the `gkgsearch` function to find knowledge about a specific entity.

3. You can use the Wikipedia search by calling the `wikisearch` function to find knowledge of a term, concept, or topic.

4. You can use the database by calling the `askdb` function to find theoretical support from encyclopedias, textbooks, and research papers.

> 5. You can use the `probe` function to find historical time series and related information, the `probe` function accepts ICode with the format "[DOMAIN]:[CODE]", for example, the Apple company stock price is "FIN:AAPL", FIN is the domain, AAPL is the code. The ICode can be queried by calling the `query_icode` function, you should never make up an ICode. There are five domains, "FIN", "WEB", "FTE", "FRD", "YGV", interpretations for them:
>
> > a) FIN: Close price time series of a financial instrument including stocks, ETFs, index funds, REITs, futures, currencies, indices, or cryptocurrencies.
> >
> > b) WEB: The Google trend time series of a keyword, such as "Apple", "Trump", "Bitcoin", etc.
> >
> > c) FTE: Political poll tracking time series, such as the president's approval rating, the generic ballot, etc.
> >
> > d) FRD: Economic time series, such as GDP, PPI, unemployment rate, etc.
> >
> > e) YGV: Public opinion poll tracking time series, such as support for universal health care, how sustainability policies impact Americans' purchase behavior, etc.

## F.4 ACTUATOR

Context of the actuator. It will be inputted to the actuator agent as the second system message after the hyperportfolio context.

> **Your role in the team is the actuator that manages the hyperportfolio of the team:**
>
> 1. You need to use your strategy to adjust and optimize your team's portfolio by making buy, sell, query, or wait for decisions based on the analysis of the latest news and information provided by the analyst, and the current status of the account which includes the holdings of assets and the excess cash.
>
> 2. In order to buy or sell assets, you need to call the `trade` function, you should pass the trading instructions to the function:
>
> > a) You should write one instruction per line.
> >
> > b) There are two kinds of instruction: BUY [ICode] [amount], SELL [ICode] [amount], all instructions should follow this format.
> >
> > c) For example, if you wish to buy ICode FIN:AAPL with 2000 amount of cash, then sell 8000 amount of FIN:GOOG, the instructions should be: BUY FIN:AAPL 2000\nSELL FIN:GOOG 8000
>
> 3. When you decide to buy or sell an asset, you usually first need to query the ICodes of the related asset by calling the `query` function unless you already know the ICode from the context.
>
> 4. The `query` function will return the ICodes for the best match assets after execution, then you need to decide the amount and pass an ICode and amount to call the `trade` function.
>
> 5. If there are no assets you want from the query, you can query again with a different query content.
>
> 6. You need to make your judgments based on the analysis, your action history and account status, and make function calls about what to do next, such as query related assets, buy or sell an asset, or wait.
>
> 7. You should always make one function call in your response.
>
> 8. You can use the `probe` function to obtain historical time series and related information to help you make your decision. The `probe` function accepts ICode as input, and ICode can be queried by calling the `query` function. You should never make up an ICode.
>
> **Here are some tips for making the decision:**
>
> 1. The asset you buy should have a valid ICode you know of or from the query.
>
> 2. The amount should always be positive.

3. The asset you sell must be the one you own, and the amount should be less than its current value.

4. Use your cash wisely, invest more if you see a good opportunity and you have confidence, invest less when you do not confirm, and learn to leave some cash for future opportunities, when you see a risk, sell the assets, the higher risk, the more you sell.

5. Higher profit opportunities always come with higher risk, which may cause a higher loss.

6. Here are some empirical principles when you decide the amount to buy or sell:

a) Usually if the asset value occupies less than 5% of your total asset, it is low and low risk.

b) The asset value that occupies between 5%~25% of your total asset will give you moderate risk.

c) The asset value that occupies between 25%~50% of your total asset will give you high risk.

### F.5 SYSTEM MESSAGES

In addition to the prompts that tell the agent their role and context, there are system messages that guide their work process.

**State messages.** When the analyst receives the latest news, article, etc., the system will inform it current time of the world, for example:

[**System**] Current time is 2021-10-02 04:01:00+00:00.

The actuator is also informed by the system about the account status including the hyperportfolio details at the beginning and after trading instructions are executed:

[**System**] Current status of the account: {account status}

**Error handling.** The analyst, assistant, and actuator all work in a loop manner, every step needs to call a function, if the agent does not call a function, the system will notify it, taking the analyst as an example:

[**System**] You do not call any function. You must call one of `ask` or `done`.

There are also other kinds of errors, such as the wrong function calling, or wrong action (e.g. the value of a trading instruction is negative), there will be a system message similar to:

[**System**] You caused an error {error message}. Please, retry.

**Final response.** When the analyst calls the `done` function, the system will prompt the analyst to provide the final full analyst report:

[**System**] Now, give the final full analysis report of the news, article, etc. based on your dialog with the `ask` function, you should use the returned evidence and information and make explicit references; you should leave a reference section with the sources listed there unless you have no

> reference. You should also include a summary of the given news, article, etc. at the beginning of your report to give the context of your analysis, at the end of your report, you should give some advice to the actuator. You should never make up any facts, evidence, or information.

For Hypothesis & Proof analyst, there is an additional sentence in the prompt:

> You should clearly state your hypothesis and proof and your intuitions.

When the assistant call `done`, it will be prompted:

> [**System**] Now, based on the search results, give your final reply to the query from the analyst "{query}". You must clearly reference the search results returned by the function calls.

**End of step.** The system will prompt the agent what to do next after successfully executing an action after each step, for example:

> [**System**] Now you can stop and wait for more news and opportunities, or you can continue to give more buy or sell instructions based on the analysis, current status of the account, and your action history.
>
> [**System**] You may choose to buy an asset from them or sell an asset you own based on the query result, the current status of the account and the analysis, or perform another query, or wait.
>
> [**System**] Now based on the results, accomplish the query. If there is an error such as an incorrect ICode when calling the probe, handle it, and retry. You can also perform a further search or decompose the query. When you finish, call `done` function.

## G    EXAMPLE OF HYPOTHESIS & PROOF ANALYST AND COMPARISON WITH PLAN & SOLVE

We demonstrate the full analysis process of our proposed Hypothesis & Proof (H&P) prompting and the Plan & Solve (P&S) prompting, the state-of-the-art prompting technique that can be applied to the analyst. Both methods are given the same input as shown in G.1. The H&P analyst then makes three queries with increasingly complex hypotheses to the assistant, as shown in G.2.1, while the P&S analyst makes only one fact-related query presented in G.2.2. Finally, the H&P and P&S analysts generate the final analysis report as seen in G.3.1 and G.3.2 respectively.

### G.1    INPUT MESSAGE

> **Current time is 2021-10-08 23:49:00+00:00, here is the latest news:**
>
> **Category**: TECH
>
> **Headline**: Apple Filing Notice of Appeal in Epic Antitrust Case, Looks to Stay In-App Injunction
>
> **Subheadline**: The iPhone maker wants a stay on judge's order to open in-app communications that could allow cheaper payment options
>
> **Datetime**: 2021-october-8 7:49 PM ET
>
> Told a federal judge on Friday it plans to appeal a verdict in its high-profile antitrust case against Epic Games Inc., a contest that has brought new attention to how big technology

companies manage software and applications on their platforms. The iPhone maker also is seeking to put on hold an order by U.S. District Judge Yvonne Gonzalez Rogers that requires Apple to loosen some restrictions on how App Store developers seek payments for their online offerings. Epic, the maker of the popular game "Fortnite," already had filed a notice of appeal to the Sept. 10 verdict issued by the Oakland, Calif., federal court following the 16-day bench trial in May. An Epic spokeswoman declined to comment on Apple's plan to appeal. The case put on trial the ways in which companies such as Apple control access to their users. Lawmakers and regulators in the U.S., European Union and beyond are closely monitoring the issue. Apple faced a deadline to file its notice of appeal, a routine procedure that doesn't include details of its argument against the one of 10 claims that the judge ruled on. Most of Epic's claims were rejected by the judge, including that Apple was an improper monopoly in its control of distribution of software onto the iPhone and its requirement that forces developers to use its in-app payment system. Apple collects what Epic calls an unfair commission of as high as 30%. Apple had argued its rules protected users and its fees were in line with industry norms and fair for the services it provides. Following the original verdict, Apple has cheered the findings as a "resounding victory". Still, it wasn't a complete win. The judge ruled that Apple was wrong to prohibit developers from using in-app and out-of-app communications with users to send them to alternative payment methods. She issued an injunction to force Apple to stop this ban.

Apple wants a stay on the injunction while it works on a solution. The company already had agreed to a settlement in another case to allow out-of-app communications, but it remains unclear how Apple wants to address in-app communications. "The company understands and respects the Court's concerns regarding communications between developers and consumers," the company said in a court filing. "Apple is carefully working through many complex issues across a global landscape, seeking to enhance information flow while protecting both the efficient functioning of the App Store and the security and privacy of Apple's customers". Some legal observers have said they think an appeal to the Ninth U.S. Circuit Court of Appeals could take a year or more, and the case is likely to ultimately go to the U.S. Supreme Court, the nation's top court. Epic co-founder Tim Sweeney has suggested the process could take five years. Meanwhile, Apple isn't allowing Epic back on its developer platform until the appeals process is completed. The legal battle emerged in August 2020, when Epic sneaked an in-app payment into its popular "Fortnite" videogame that violated Apple's rules. In response, Apple booted the game from the App Store, and Epic's lawsuit followed. The case gave a unique window into Apple's secretive App Store business, revealing that about one fifth of operating income in fiscal 2019 came from its cut of app revenue, according to a Wall Street Journal analysis of court records that were made public as part of the trial. Apple has said that the released figures were flawed and too high; still, it has acknowledged that much of its App Store revenue came from videogames.

**Now, give your analysis that can help the actuator optimize the hyperportfolio of the team by buying or selling assets for this news.**

## G.2 ANALYSIS PROCESS

### G.2.1 HYPOTHESIS & PROOF

The first `ask` called by the Hypothesis & Proof analyst:

**[Analyst]** Ask assistant: **What is the potential impact of Apple filing notice of appeal in the Epic antitrust case?**

**[Assistant]** Based on the search results, Apple has filed a notice of appeal in the Epic Games antitrust case. The filing of this appeal could potentially delay any changes to the App Store rules that were ordered by the court. According to CNBC, in September, federal judge Yvonne Gonzalez Rogers ruled in favor of Apple for nine out of ten counts in the antitrust trial brought by Epic Games, the maker of Fortnite (source: [CNBC](https://www.cnbc.com/2021/10/08/apple-files-appeal-in-epic-games-case-that-could-delay-changes-.html)). This ruling was seen as a resounding victory for Apple in the case. Bloomberg Law also reports that Epic Games has filed a notice of appeal, indicating their intention to challenge the court's ruling (source: [Bloomberg Law](https://news.bloomberglaw.com/antitrust/epic-games-v-apple-ruling-antitrust-litigation-takeaways)).

While there is no specific information available about the potential impact of the appeal, it is clear that both Apple and Epic Games are actively involved in the legal proceedings and are seeking resolution in their favor. Further monitoring of this case and any subsequent court decisions or actions would provide more insights into the potential impact of Apple filing a notice of appeal in the Epic antitrust case.

[**Analyst**] Based on the news that Apple has filed a notice of appeal in the Epic Games antitrust case, it indicates that the legal battle between Apple and Epic Games is ongoing and that Apple is seeking to challenge the court's ruling. This development suggests that there may be potential implications for Apple's business practices and the App Store's rules in the future.

As an analyst, it is challenging to determine the exact impact of this appeal on Apple's stock price or the overall financial market. However, it is worth noting that legal proceedings and regulatory actions can have a significant impact on the performance of companies in the technology sector. Investors and market participants may closely monitor the developments in this case to assess the potential risks and opportunities associated with Apple's business.

Given the uncertainty surrounding the outcome of the appeal and its potential impact on Apple, it may be prudent for the actuator to closely monitor the progress of the case and assess any updates or rulings that may arise in the future. The actuator can consider incorporating this information into their decision-making process, especially if they have a significant exposure to Apple stock or related assets in the hyperportfolio.

Ultimately, the decision to buy or sell assets related to Apple or adjust the hyperportfolio should be based on a comprehensive evaluation of various factors, including the outcome of the appeal, market conditions, and the actuator's risk tolerance and investment strategy.

The second `ask` called by the Hypothesis & Proof analyst:

[**Analyst**] Ask assistant: **Has there been any significant market reaction to the news of Apple filing an appeal in the Epic Games antitrust case?**

[**Assistant**] Based on the search results, there is no direct evidence of any significant market reaction to the news of Apple filing an appeal in the Epic Games antitrust case. However, it is important to note that market reactions can be complex and influenced by various factors. To get a comprehensive understanding of any market reactions, it would be beneficial to analyze the stock price of Apple (AAPL) over time and compare it to the timeline of the news and events related to the Epic Games antitrust case. Additionally, monitoring the news and expert opinions related to Apple's legal battles and antitrust concerns can provide further insights into potential market reactions.

[**Analyst**] Based on the information available, there is no direct evidence to indicate a specific market reaction to Apple filing an appeal in the Epic Games antitrust case. However, it is important to note that market reactions can be influenced by a multitude of factors and may not always be immediate or easily observable. News and legal developments related to major companies like Apple can have an impact on investor sentiment and market dynamics over time.

To gain a better understanding of the potential market reaction, it may be useful to monitor any updates or developments related to the court case, as well as any commentary or analysis from industry experts and market observers. This ongoing analysis can help inform investment decisions and portfolio management strategies.

It is also important to consider that individual stock movements are influenced by various factors, including company-specific news, broader market conditions, and investor sentiment. Therefore, it is crucial to conduct comprehensive research and analysis when making investment decisions and managing a hyperportfolio.

As an analyst, it can be helpful to keep track of any new information and developments related to the Apple-Epic Games case and its potential impact on market sentiment and Apple's business operations. This ongoing monitoring can provide valuable insights for optimizing the hyperportfolio.

The third `ask` called by the Hypothesis & Proof analyst:

[Analyst] Ask assistant: **Is there any evidence of market reactions to legal proceedings or appeals in similar antitrust cases involving large technology companies?**

[Assistant] Based on the search results, I found information regarding antitrust cases involving large technology companies, specifically the case against Microsoft Corporation in 1998. In this case, Microsoft was accused of using its significant market share in the personal computer operating systems business to prevent competition and maintain a monopoly. The case resulted in a federal district judge ruling that Microsoft should be broken into two technology companies, but this decision was later reversed on appeal by a higher court.

Unfortunately, I couldn't find specific market reactions to legal proceedings or appeals in similar antitrust cases involving large technology companies in the search results provided. It's possible that more specific information or analysis on market reactions in such cases may require further research or analysis.

Please note that the search results provided are limited and may not encompass all relevant information. Further research or analysis may be necessary to gather comprehensive evidence on market reactions to legal proceedings or appeals in similar antitrust cases involving large technology companies.

[Analyst] Based on the available information, there is limited evidence regarding the market reactions to legal proceedings or appeals in similar antitrust cases involving large technology companies. The search results did not yield specific information or analysis on market reactions to legal proceedings or appeals in cases such as the Microsoft antitrust case in 1998.

However, it is important to note that market reactions to legal proceedings can be complex and influenced by various factors, including the specific details of the case, the potential impact on the company's operations and profitability, investor sentiment, and broader market conditions. Market reactions may also vary depending on the outcome or ruling of the legal proceedings.

To gain a comprehensive understanding of potential market reactions to legal proceedings or appeals in antitrust cases involving large technology companies, it may be beneficial to conduct further research and analysis, including examining historical precedent or consulting industry experts and market analysts.

In the context of Apple filing an appeal in the Epic Games antitrust case, monitoring future developments in the case, as well as any updates or expert opinions, may provide insights into potential market reactions. This information can help inform the actuator's decision-making process regarding the hyperportfolio and the management of Apple-related assets.

Ultimately, investment decisions should be based on a thorough analysis of various factors, including legal proceedings, market conditions, company performance, and risk tolerance. Regular monitoring and evaluation of relevant news and events can help optimize the hyperportfolio and respond to potential market movements.

**I'm ready to give the final analysis report now.**

### G.2.2 PLAN & SOLVE

The only `ask` called by the Hypothesis & Proof analyst:

[Analyst] Ask assistant: **What is the Epic antitrust case against Apple about?**

[Assistant] The Epic antitrust case against Apple, officially known as Epic Games v. Apple, is a lawsuit brought by Epic Games against Apple in August 2020. The case was filed in the United States District Court for the Northern District of California. According to the search results from the wikisearch, Epic Games filed the lawsuit against Apple for antitrust violations and anti-competitive behavior, along with a similar lawsuit against Google. The lawsuit alleges that Apple's app store practices violate antitrust laws. More information about the case can be found in the search results.

[Analyst] Thank you for the information. Based on the news article, it seems that Apple is planning to appeal the verdict in the Epic antitrust case. The article mentions that Apple

wants to stay on the judge's order to open in-app communications that could allow cheaper payment options. The case focuses on how big technology companies, like Apple, control access to their platforms and the restrictions they impose on app developers.

From an investment perspective, this news may have implications for Apple's business operations and its relationship with app developers. It is worth monitoring how this legal battle unfolds and any potential changes in Apple's App Store policies. Any significant developments in the case could impact Apple's reputation and potentially its stock performance.

At this point, it does not provide a clear indication to buy or sell Apple stock. However, it is essential to keep an eye on any updates regarding the case and how it might impact Apple's financials and market position.

**I'm ready to give the final analysis now.**

## G.3 ANALYSIS REPORT

### G.3.1 HYPOTHESIS & PROOF

**Summary**

The news reveals that Apple has filed a notice of appeal in the Epic Games antitrust case. This appeal comes after a high-profile trial where a federal judge ruled in favor of Apple on most counts but issued an injunction requiring Apple to loosen some restrictions on how App Store developers seek payments. Both Apple and Epic Games have filed notices of appeal, indicating their intentions to challenge the court's ruling. While the news does not provide specific details about the potential impact of the appeal, it highlights the ongoing legal battle between Apple and Epic Games and the implications for the App Store's rules and business practices.

**Hypothesis**

1. The appeal in the Epic Games antitrust case could lead to potential changes in Apple's App Store rules and business practices.

2. The outcome of the appeal could have implications for Apple's financial performance and its stock price.

**Proof**

1. The news confirms that Apple has filed a notice of appeal in the antitrust case against Epic Games, indicating their intention to challenge the court's ruling and potential changes to the App Store's rules.

2. The trial and the subsequent appeal have drawn significant attention from lawmakers, regulators, and market participants, highlighting the potential impact on Apple's business practices.

**Intuitions**

1. The ongoing legal battle and uncertain outcome of the appeal may create volatility and uncertainty for Apple and its stakeholders, including investors.

2. Market reactions to legal proceedings and appeals in high-profile antitrust cases involving technology companies can be complex and influenced by various factors such as investor sentiment, broader market conditions, and the specific details of the case.

**Advice**

1. The actuator should closely monitor the progress of the appeal and any updates or rulings related to the case.

2. Considering the potential impact on Apple's business practices, it is important for the actuator to assess the potential risks and opportunities associated with Apple's stock and related assets in the hyperportfolio.

3. Stay informed about the developments in the Epic Games antitrust case and seek expert opinions and analysis to make informed decisions regarding the hyperportfolio, taking into account factors such as the outcome of the appeal and market conditions.

### References

- WSJ: "Apple Filing Notice of Appeal in Epic Antitrust Case, Looks to Stay In-App Injunction" (Author: Tim Higgins)

- CNBC: "Apple files appeal in Epic Games case that could delay changes" (Oct 8, 2021)

- Bloomberg Law: "Epic Games v. Apple Ruling: Antitrust Litigation Takeaways" (source: Bloomberg Law)

## G.3.2    PLAN & SOLVE

### Summary

The news article discusses Apple's plan to appeal the verdict in the Epic antitrust case. The case revolves around Apple's App Store practices and restrictions imposed on app developers. Apple is seeking to stay a judge's order to open in-app communications that could allow cheaper payment options for developers. This legal battle has drawn attention to how big technology companies control access to their platforms.

### Analysis

1. Legal Implications: The outcome of the Epic antitrust case will have legal implications for Apple. If Apple's appeal is successful, it could reinforce the company's control over its App Store and the restrictions it imposes. On the other hand, if the appeal is denied, Apple may be required to make changes to its App Store policies, potentially impacting its revenue streams.

2. App Store Business Model: Apple's App Store is a significant source of revenue for the company. The article mentions that approximately one-fifth of Apple's operating income in fiscal 2019 came from its cut of app revenue. Any changes in App Store policies or regulations could have financial implications for Apple.

3. Developer Relations: The outcome of the case could also impact Apple's relationship with app developers. If Apple is required to allow cheaper payment options, it may affect the revenue generated by in-app purchases and subscriptions. Developers may have more flexibility and potentially lower costs, which could impact their willingness to continue developing for Apple's platform.

### Recommendation

Based on the current information available, it is difficult to determine the impact of the Epic antitrust case on Apple's stock price or financial performance. It would be prudent to closely monitor any developments in the case and the reactions of the market and industry. Additionally, keeping an eye on any potential changes in Apple's App Store policies and their implications for developers can provide insight into future opportunities or risks.

### References

- WSJ article: [insert link]

- Summary of Epic antitrust case against Apple: [insert reference]

- Impact of App Store revenue on Apple's financials: [insert reference]

# H  EXAMPLES OF ASSISTANT

We show the process of how the assistant handles the three queries made by the H & P analyst in the example in Appendix G here.

The process of the first query from the H&P analyst:

---

**[Analyst] What is the potential impact of Apple filing notice of appeal in the Epic antitrust case?**

[Assistant] **Search_google("potential impact of Apple filing notice of appeal in the Epic antitrust case")**

[Function] Return for your query potential impact of Apple filing notice of appeal in the Epic antitrust case:

```
{'top 1 result':  {'url':  'https://www.cnbc.com/2021/10/08/
apple-files-appeal-in- epic-games- case-that-could-
delay-changes-.html', 'title':  'Apple files appeal in Epic Games
case that could delay ...', 'description':  'Oct 8, 2021 -- In
September, federal judge Yvonne Gonzalez Rogers ruled in favor of
Apple for nine of 10 counts in an antitrust trial brought by Epic,
the maker of Fortnite.'}, 'top 2 result':  ...  (omitted here)}
```

[Assistant] Based on the search results, Apple has filed a notice of appeal in the Epic Games antitrust case. The filing of this appeal could potentially delay any changes to the App Store rules that were ordered by the court. According to CNBC, in September, federal judge Yvonne Gonzalez Rogers ruled in favor of Apple for nine out of ten counts in the antitrust trial brought by Epic Games, the maker of Fortnite (source: [CNBC](https://www.cnbc.com/2021/10/08/apple-files-appeal-in-epic-games-case-that-could-delay-changes-.html)). This ruling was seen as a resounding victory for Apple in the case. Bloomberg Law also reports that Epic Games has filed a notice of appeal, indicating their intention to challenge the court's ruling (source: [Bloomberg Law](https://news.bloomberglaw.com/antitrust/ epic-games-v-apple-ruling-antitrust-litigation-takeaways)).

While there is no specific information available about the potential impact of the appeal, it is clear that both Apple and Epic Games are actively involved in the legal proceedings and are seeking resolution in their favor. Further monitoring of this case and any subsequent court decisions or actions would provide more insights into the potential impact of Apple filing a notice of appeal in the Epic antitrust case.

---

The process of the second query from the analyst:

---

**[Analyst] Has there been any significant market reaction to the news of Apple filing an appeal in the Epic Games antitrust case?**

[Assistant] **Search_google("Apple filing appeal in Epic Games antitrust case news")**

[Function] Return for your query Apple filing appeal in Epic Games antitrust case news:

```
{'top 1 result':  {'url':  'https://news.bloomberglaw.com/
antitrust/ epic-games-v- apple-ruling- antitrust-
litigation-takeaways', 'title':  'Epic Games v.  Apple Ruling:
Antitrust Litigation Takeaways', 'description':  "Oct 6, 2021 --
The federal judge overseeing the antitrust suit brought by Epic Games
against Apple mostly sided with the iPhone maker, including ruling
that Apple's App\xa0..."}, 'top 2 result':  ...  (omitted here)}
```

[Assistant] **Query_data("FIN:AAPL")**

[Function] Return for your query FIN:AAPL:

{'current': 140.3759460449,

---

'history':

datetime                                    value

2020-10-28 04:00:00+00:00 109.225708

2020-11-03 05:00:00+00:00 108.479187

... (omitted 46 lines here)

2021-09-30 04:00:00+00:00 139.881683

2021-10-06 04:00:00+00:00 140.375946,

'information': ... (omitted here)}

[Assistant] Based on the search results, there is no direct evidence of any significant market reaction to the news of Apple filing an appeal in the Epic Games antitrust case. However, it is important to note that market reactions can be complex and influenced by various factors. To get a comprehensive understanding of any market reactions, it would be beneficial to analyze the stock price of Apple (AAPL) over time and compare it to the timeline of the news and events related to the Epic Games antitrust case. Additionally, monitoring the news and expert opinions related to Apple's legal battles and antitrust concerns can provide further insights into potential market reactions.

The process of the third query from the analyst:

[Analyst] **Is there any evidence of market reactions to legal proceedings or appeals in similar antitrust cases involving large technology companies?**

[Assistant] **Query_KB("market reactions to antitrust cases involving large technology companies")**

[Function] Return for your query market reactions to antitrust cases involving large technology companies:

```
 {'ids':  [['1004962', '377417', '1874968', '154887', '53772']],
'distances':  [[0.23472154140472412, 0.2480563521385193,
0.24832972884178162, 0.25399601459503174, 0.2581632435321808]],
'documents':  [[' Microsoft wasstimulated largely by complaints
from AOL and Netscape, Microsoft's maincompetitors in Web browser
technology.\n4.The relationship between concentration and prices
is debatable.  A core tenet\nof antitrust economics is that
anticompetitive market behavior follows from in-creased concentration
in an industry.  Judge Richard Posner complained, how-ever, that
there was not enough "empirical guidance from industrialorganization
economists,"... (omitted here)}
```

[Assistant] **Query_data("FIN:MSFT")**

[Function] Return for your query FIN:MSFT:

{'current': 293.1099853516,

'history':

datetime                                    value

2020-10-28 04:00:00+00:00 201.221588

2020-11-03 05:00:00+00:00 204.944611

... (omitted 46 lines here)

2021-09-30 04:00:00+00:00 281.920013

2021-10-06 04:00:00+00:00 293.109985,

'information': ... (omitted here)}

[Assistant] Based on the search results, I found information regarding antitrust cases involving large technology companies, specifically the case against Microsoft Corporation in 1998. In this case, Microsoft was accused of using its significant market share in the personal computer operating systems business to prevent competition and maintain a monopoly. The case resulted in a federal district judge ruling that Microsoft should be broken into two technology companies, but this decision was later reversed on appeal by a higher court.

Unfortunately, I couldn't find specific market reactions to legal proceedings or appeals in similar antitrust cases involving large technology companies in the search results provided. It's possible that more specific information or analysis on market reactions in such cases may require further research or analysis.

Please note that the search results provided are limited and may not encompass all relevant information. Further research or analysis may be necessary to gather comprehensive evidence on market reactions to legal proceedings or appeals in similar antitrust cases involving large technology companies.

## I  EXAMPLE OF ACTUATOR

### I.1  INPUT

**This is the analysis report of the latest news, article, etc. from the analyst:**

**Context:**

The given news article highlights the impact of the Apple App Store on Apple Inc.'s business and profitability. It emphasizes the central role of the App Store in driving Apple's profits, particularly in the gaming sector. The article discusses the App Store's dominance in the gaming economy, revenue generated from games, potential threats to Apple's position, and the increasing scrutiny from lawmakers and regulators.

**Hypothesis:**

The news about the App Store's impact on Apple's business will have a positive effect on the stock price of Apple Inc. (AAPL).

**Evidence:**

1. The growth of the App Store: The news mentions that the App Store has experienced significant growth, reaching 2 million apps by 2018 (Apple Insider). This growth indicates the increasing popularity of apps and suggests a positive impact on Apple's business, which could potentially affect the stock price of AAPL.

2. Significance of the App Store: The article emphasizes the dominant role of the App Store in the gaming economy, where Apple takes a 30% cut of sales from thousands of games. It states that Apple raked in more profits from games than gaming giants Microsoft, Nintendo, and Sony combined (WSJ). This reinforces the importance of the App Store in Apple's business model and implies a potential positive impact on AAPL's stock price.

**Proof and Intuition:**

Based on the evidence provided, including the growth of the App Store and its significance in Apple's business, it is reasonable to infer that the news about the App Store's impact will likely have a positive effect on the stock price of AAPL. The strong revenue generated from games and Apple's dominance in the gaming economy further support this hypothesis.

However, it is important to consider that stock prices are influenced by various factors, and the relationship between news and stock prices can be complex. Other factors such as overall market conditions, financial performance, product launches, and investor sentiment should also be considered for a comprehensive analysis.

**Summary:**

The news article highlights the App Store's crucial role in driving Apple's profits, particularly in the gaming sector. It emphasizes Apple's dominance, revenue generated from games, and the potential threats and challenges facing the company. The growth and significance of the

App Store suggest a positive impact on Apple's business and potentially on the stock price of AAPL.

**Advice:**

Based on the analysis, it would be prudent for the actuator to consider the positive impact of the App Store on Apple's business and its potential influence on the stock price of AAPL. However, it is recommended to conduct further analysis, consider other factors that influence stock prices, and monitor market conditions for a holistic understanding of AAPL's performance.

**Current status of the account:**

```
{'account': {'account name': 'default', 'cash': 989900.0,
'cash ratio': 0.99, 'assets value': 10000.0, 'assets ratio':
0.01, 'total value': 999900.0, 'portfolios': {'default':
{'total': 10000.0, 'FIN:HCA': {'symbol': 'FIN:HCA', 'name':
'HCA Healthcare Inc. Common Stock', 'value': 10000.0, 'price':
251.7129516602, 'unrealize': 0.0, 'base': 10000.0, 'size':
19.863896, 'overnight_total': 0.0, 'ratio': 0.01}, 'recent
transactions':  ...  (omitted here)}
```

**Now, do your judgments, make decisions, and do function callings. Remember to include the reasoning steps in your judgments and decision-making process.**

## I.2 ACTIONS

[Actuator] **Query_icode("Apple stock")**

[Function] Here are the top 3 candidates for the query "Apple stock":

1. Code: FIN:AAPL (distance 0.22227655351161957)

This item is the close price time series of a financial instrument including stocks, ETFs, index funds, REITs, futures, currencies, indices, or cryptocurrencies, information about this financial instrument is listed below:

- Name: Apple Inc. Common Stock

... (omitted here)

[Actuator] Based on the analysis report and the current status of the account, it is recommended to hold the current asset, FIN:AAPL (Apple Inc. Common Stock), as the news article suggests a positive impact on Apple's business and potentially on the stock price of AAPL. The analysis highlights the growth of the App Store, its significance in Apple's business, and the strong revenue generated from games.

Considering the current cash ratio of 0.99, it would be wise to allocate some additional cash to increase the position in FIN:AAPL. However, we need to ensure that the asset value represented by FIN:AAPL does not exceed the risk level we are comfortable with. As per the empirical principles mentioned earlier, having the asset value between 5% to 25% of the total portfolio is considered a moderate risk.

You may allocate a portion of the remaining cash to buy more of FIN:AAPL, but be mindful of the commission cost and the risk associated with concentrating too much in a single asset. It is important to maintain a diversified portfolio to mitigate risk.

[Actuator] **Trade("BUY FIN:AAPL 49900")**

[Function] Succeed. All instructions are executed. After executing the instruction, the current status of the account ... (omitted here)

[Actuator] Given the current status of the account and the recent purchase of additional shares of FIN:AAPL, it is recommended to continue monitoring market conditions and news related to Apple Inc. and the App Store for any potential changes that may affect the stock price of AAPL.

[Actuator] **Query_icode("Google search trends for 'iPhone'")**

[**Function**] Here are the top 3 candidates for the query "Google search trends for 'iPhone'":

1. Code: WEB:SWTUI (distance 0.23514001071453094)

This item is a Google trend time series that reveals the popularity of a keyword in internet search, information about this keyword is listed below:

- Keyword: Swiftui

- Description: Toolset for building user interfaces for Apple platforms. It simplifies the coding process and provides a side-by-side preview.

- Category: Software, Technology, Design

- Related topics: Server Side Tracking, User Interface Design, User Engagement, User Interviews

... (omitted here)

[**Actuator**] Based on the information provided, the top candidates for the query "Google search trends for 'iPhone'" are not directly related to the Apple Inc.'s iPhone product. The search trends are focused on other keywords such as "SwiftUI," "Mobile Developer," and "Mobile Revenue." These keywords may not provide direct insights into the popularity or demand for the iPhone.

Considering the current status of the account and the information available, it is recommended to wait for more relevant information or opportunities before making any further decisions regarding asset purchases or sales.

