# OpenReview forum: "SocioDojo: Building Lifelong Analytical Agents with Real-world Text and Time Series"
_ICLR.cc/2024/Conference — ICLR 2024 spotlight_

### Official Review · Reviewer_qsUb · 2023-10-30

**Soundness:** 3 good
**Presentation:** 4 excellent
**Contribution:** 3 good
**Rating:** 8
**Confidence:** 3

**Summary:**

This work tries to propose an open-ended lifelong learning environment designed for developing autonomous agents that can perform human-like and up-to-date analysis and decision-making on societal topics. To do so, the environment SocioDojo is structured to emulate the historical development of society using real-world texts and time series data.

Specifically, it 1) integrates a continuous stream of real-world texts and messages from various information sources, including news, social media, reports, and more. The information in the form of time-stamped messages, ensures that the agents are exposed to up-to-date societal developments, forcing them to analyze and respond to fresh data.
2) incorporates a vast array of time series data spanning various societal aspects like finance, economics, politics, and social trends.
3) incorporates multiple knowledge bases and tools, including books, journals,  encyclopedias, interfaces to search engines and knowledge graphs, etc.

With the environment, the task of the agent is called "Hyperportfolio Task". They are given an initial "cash" value and are tasked with making "investments" in different "assets" that correspond to various time series. The goal is to maximize their total assets over a specified period, just like investment and asset management in the real world.

In this work, the author also proposes the Analyst-Assistant-Actuator agent architecture to tackle the hyperportfolio task, and the Hypothesis and proof prompting technique for generating high-quality analysis, which achieves improvements of 32.4% and 30.4% in two experiment settings compared to the state-of-the-art methods. The ablation study results show 1)the importance of domain-specific analysis techniques and high-quality information sources. 2)The analyst is critical for the hyperportfolio task.

**Strengths:**

Overall, this paper could be a significant contribution to the research question of "How can we get an environment designed for developing autonomous agents that can perform human-like and up-to-date analysis and decision-making on societal topics."
The writing is clear and the paper is easy to follow.
The proposed approach to using diverse and real-world information sources to simulate the societal environment can truly grapple with the complexities and nuances of real-world information, which facilitates the exploration of general intelligence.

The introduction of the hyperportfolio task is novel to me, I do not think such kind of automatic metric is the best way to evaluate agents's foresight and strategic thinking ability but it is good enough.

The setting of the environment is also meaningful. Such as the prohibition of "day trading"  ensures that agents don't exploit short-term fluctuations but instead focus on understanding and predicting more meaningful, long-term societal trends.

Overall, I do believe the complexity of the hyperportfolio task could set a high bar for LLM-based agents, challenging the community to rise to the occasion.

**Weaknesses:**

While the SocioDojo environment is comprehensive and emulates the running of the world, its complexity might make it challenging for researchers to quickly adapt.

I think it would be beneficial to introduce some actual real-world classic scenarios or investment cases. These could serve as short-term goals or benchmarks, allowing researchers to run and evaluate their own model in a phased manner. It could also be useful for evaluating the information coverage level of the system.

The lower bound of the system is not clear. The complexity of SocioDojo might inadvertently obscure the foundational or simpler strategies that also be effective. For instance, SOTA time-series forecasting methods could be employed as a foundational strategy for investment within the SocioDojo environment. It could serve as a baseline or a lower bound against which more complex strategies can be compared.

I also hope the author put more discussion on how the authors have addressed potential biases, screened sources, and ensured the diversity of data, it would bolster the paper's credibility and address potential concerns.

**Questions:**

Are there any data leakage between the knowledge base and the streaming messages?

---

> ### Author Response · Authors · 2023-11-20
> **Response to Reviewer qsUb**
>
> We deeply appreciate your insightful feedback and recognition of the significance of our work, SocioDojo. We have taken your comments into careful consideration and offer the following detailed responses:
>
>
> ## For the Weaknesses:
>
>
> > 1. While the SocioDojo environment is comprehensive and emulates the running of the world, its complexity might make it challenging for researchers to quickly adapt.
>
> We acknowledge that the comprehensive nature of SocioDojo presents a learning curve. To provide some help to assist researchers in navigating this complexity, we have included a new figure in Appendix D. This figure provides an overview of the collaborative decision-making process involving the Analyst, Assistant, and Actuator, offering a clearer understanding of their roles and interactions within SocioDojo.
>
>
> > 2. I think it would be beneficial to introduce some actual real-world classic scenarios or investment cases. These could serve as short-term goals or benchmarks, allowing researchers to run and evaluate their own model in a phased manner. It could also be useful for evaluating the information coverage level of the system.
>
> Introducing real-world scenarios and investment cases is indeed a valuable suggestion. We plan to develop and include these examples in SocioDojo to serve as benchmarks and to enhance the system's evaluation in terms of information coverage and real-world applicability.
>
>
> > 3. The lower bound of the system is not clear. The complexity of SocioDojo might inadvertently obscure the foundational or simpler strategies that also be effective. For instance, SOTA time-series forecasting methods could be employed as a foundational strategy for investment within the SocioDojo environment. It could serve as a baseline or a lower bound against which more complex strategies can be compared.
>
> While SOTA domain-specific models in areas like quant finance could provide a foundational strategy within SocioDojo, our focus is on creating fair baselines with autonomous agents geared toward general socio-intelligence. However, incorporating expert domain models and tools into the agent's analysis process presents an exciting future direction, leveraging the capabilities of large language models (LLMs) for tool utilization.
>
>
> > 4. I also hope the author put more discussion on how the authors have addressed potential biases, screened sources, and ensured the diversity of data, it would bolster the paper's credibility and address potential concerns.
>
>
> To address potential biases and source diversity concerns, we have added Appendix B, detailing our methodologies for screening sources and ensuring data diversity.
>
>
> ## For your Questions:
>
>
> > 1. Are there any data leakage between the knowledge base and the streaming messages?
>
>
> To prevent data leakage, we have strictly used only materials published before the start date in the knowledge base.
>
> ---
>
> We hope these responses comprehensively address your points and enhance your understanding of our project. We greatly appreciate your constructive feedback and remain open to any further inquiries or suggestions for improvement. Thank you for your support and valuable contribution to our work's development.

---

### Official Review · Reviewer_jccf · 2023-10-31

**Soundness:** 3 good
**Presentation:** 4 excellent
**Contribution:** 3 good
**Rating:** 8
**Confidence:** 4

**Summary:**

This paper introduced SocioDojo, a framework for developing ready-to-deploy autonomous agents capable of performing human-like analysis and decision-making on societal topics, e.g., finance. The authors demonstrated the use of SocioDojo by a task called "hyperportfolio", where agents read news and time series data and make decision on buying/selling assets. The results show that the proposed method achieves improvements of 32.4% and 30.4% compared to the state-of-the-art method in the experimental settings of "standard" and "tracking".

**Strengths:**

* The paper was well-written with sufficient details on data sources, work process. Ablation studies explored the factors that impact performance.

* The task of hyperportfolio is carefully designed, for example, commission fee is considered.

* The proposed Analyst-Assistant-Actuator (AAA) agent architecture outperformed several recent baselines such as Self-Ask and AutoGPT.

**Weaknesses:**

* Since GPT-3.5-Turbo was used as foundation models with a non-zero (0.2) temperature, the results should not be fully deterministic, while the paper may miss some studies on randomness.

* The setting of "Forbid day trading", that an asset cannot be sold within 5 days of purchase to avoid profiting from short-term patterns, might be overly strict.

* For the experimental setting of "tracking", it remains unclear how the portfolio performs against an actual index tracker.

**Questions:**

* When multiple news articles sent out on the same day, did the order of news matter? i.e., to which extend the agent is permutation invariant?

---

> ### Author Response · Authors · 2023-11-20
> **Response to Reviewer jccf**
>
> We greatly appreciate your thoughtful feedback and acknowledgment of our work on SocioDojo. After careful consideration of your comments, we offer the following responses:
>
>
> ## For the Weaknesses:
>
> > 1. Since GPT-3.5-Turbo was used as foundation models with a non-zero (0.2) temperature, the results should not be fully deterministic, while the paper may miss some studies on randomness.
>
> We choose a non-zero setting as 0 temperature may restrict the model performance. While the chosen temperature of 0.2 (on a scale of 0 to 2) is relatively low and aligns with recommendations for deterministic tasks, we recognize the value of further exploring how varying temperatures might affect outcomes. We are currently conducting additional studies on this aspect and plan to include these findings in future updates of our work.
>
>
> > 2. The setting of "Forbid day trading", that an asset cannot be sold within 5 days of purchase to avoid profiting from short-term patterns, might be overly strict.
>
> The stringent "Forbid day trading" rule was designed to elevate the cost of decision-making, thereby promoting depth and consideration in the agents' analyses. This aligns with our goal of fostering high-quality, thoughtful decision-making. Additionally, our use of daily close price data, which are public while minute-level data are usually commercial, necessitates such a restriction to counterbalance the potential information advantage that agents might have due to data delays. We have added Appendix D to examine the implications of utilizing close price data.
>
>
> > 3. For the experimental setting of "tracking", it remains unclear how the portfolio performs against an actual index tracker.
>
> In response to your query about portfolio performance in the "tracking" experimental setting, we have added a comparison with major indices like S&P 500 and FANG+ in Appendix C. This comparison helps contextualize our agent's performance, although it's important to note that SocioDojo has not been explicitly optimized for financial market environments. The direct comparison may not be fair considering the agent's unique advantages and disadvantages, such as not adhering to trading hours and not utilizing real-time data, along with the restriction against day trading.
>
>
>
> ## For your Questions:
>
> > 1. When multiple news articles sent out on the same day, did the order of news matter? i.e., to which extend the agent is permutation invariant?
>
> In SocioDojo, each news piece is timestamped to maintain chronological order, mirroring real-world scenarios. Overlaps in timestamps are rare, but even if they occur, they accurately reflect real-life information flows, ensuring that the agent's response is realistically grounded.
>
> ---
>
> We hope these responses address your concerns and provide further clarity on our research. We are grateful for your feedback and remain open to any additional questions or suggestions you might have. Thank you once again for your valuable input and for the opportunity to enhance our work.

---

### Official Review · Reviewer_apFV · 2023-11-01

**Soundness:** 2 fair
**Presentation:** 2 fair
**Contribution:** 2 fair
**Rating:** 5
**Confidence:** 3

**Summary:**

The paper introduces SocioDojo, a comprehensive and open-ended learning environment designed for the development of autonomous agents capable of conducting human-like analysis and decision-making in areas such as economics, finance, politics, and culture. Such quality of analysis is evaluated by a proposed “hyperportfolio” framework, which generalizes from financial time series to all different time series. The paper also introduces a novel architectural agent approach, the Analyst-Assistant-Actuator, specifically for the “hyperportfolio” management task. The result is evaluated by the “return” of the “hyperportfolio” in a given period of time and it is shown to outperform other approaches like Self-Ask and Plan & Solve.

**Strengths:**

1. SocioDojo can take in a lot of real-world information sources and knowledge base
2. Propose AAA agent architecture, with an analyst, an assistant, and an actuator, and it seems to work well in the proposed “hyperportfolio” management task.
3. The proposed “hyperportfolio” is novel and interesting, which takes in a large number of different time series and can be used to evaluate the agent’s overall understanding of various aspects of real world

**Weaknesses:**

1. The definition of POMDP seems to disconnect from the rest part of the manuscript. Those notations are not used in other parts at all.
2. The main evaluation results are only based on a final return in the defined period, which misses other important aspects for the results to be valid. See questions below
3. The proposed “return” of “hyperportfolio”, as some kind of evaluation metric, is very hard to interpret. While it seems that it is related to how accurately the agent is able to predict the future of various time series, it’s hard to make sense of the numerical value, especially when various time series are fused together.

**Questions:**

1. Instead of expected return, modern portfolio theory typically tries to maximize risk-adjusted return. Is there any specific reason that the authors do not consider the standard deviation?
2. In Table 1, it is shown that there would be 9 research papers per day. Is this an average statistic or subscribed information that will constantly have 9 papers?
3. How does the “return” of “hyperportfolio” change over time? How is the final “return” distributed across different time periods?
4. For some of the time series, e.g. GDP as an economic time series, we would have existing forecasting beforehand. They might be part of the report, and the model can directly use these numbers, which are typically fairly accurate. In this case, how do you confirm that the agent utilizes the comprehensive information from different sources to make the “investment” instead of directly taking those numbers?
5. Financial time series is typically the most noisy one compared with other time series, which should also be the least predictable. However, it is shown that the financial time series achieves the highest return. How do authors interpret such results?
6. Typos: equation 3.1.2, Table 3.2.1 and Figure 3.2.1 are not pointing to the correct place and are not indexed correctly. Table 1 in the Knowledge Base part should be Table 2.

---

> ### Author Response · Authors · 2023-11-20
> **Response to Reviewer apFV (Part 1/2)**
>
> We are grateful for your comprehensive review and constructive feedback on our work with SocioDojo. Below, we address each of your concerns and questions:
>
> ## For the Weaknesses:
>
> > 1. The definition of POMDP seems to disconnect from the rest part of the manuscript. Those notations are not used in other parts at all.
>
> We appreciate your observation regarding the use of POMDP in our manuscript. In Section 3.2, we clarify how SocioDojo implements the POMDP environment, with information sources forming the observation space $\Omega$, time series approximating the world state $S$, and the knowledge bases as context $Z$. This formulation in POMDP terms offers a structured and compact mathematical framework, enhancing the manuscript's comprehensibility and providing a common language for future research in this domain.
>
>
> > 2. The main evaluation results are only based on a final return in the defined period, which misses other important aspects for the results to be valid. See questions below
>
> Further addressed below in “Questions”.
>
> > 3. The proposed “return” of “hyperportfolio”, as some kind of evaluation metric, is very hard to interpret. While it seems that it is related to how accurately the agent is able to predict the future of various time series, it’s hard to make sense of the numerical value, especially when various time series are fused together.
>
> Further addressed below in “Questions”.

---

> > ### Author Response · Authors · 2023-11-20
> > **Response to Reviewer apFV (Part 2/2)**
> >
> > ## For the Questions:
> >
> > > 1. Instead of expected return, modern portfolio theory typically tries to maximize risk-adjusted return. Is there any specific reason that the authors do not consider the standard deviation?
> >
> > Our initial focus on expected return was for simplicity and intuitive comparison. Acknowledging your valid point, we have now included Sharpe Ratio calculations in Table 3 to provide a more comprehensive risk-adjusted performance assessment.
> >
> >
> > > 2. In Table 1, it is shown that there would be 9 research papers per day. Is this an average statistic or subscribed information that will constantly have 9 papers?
> >
> > The figure quoted is an average. We have updated the table to clarify this by changing the label to “Avg. Per Day” to prevent confusion.
> >
> >
> > > 3. How does the “return” of “hyperportfolio” change over time? How is the final “return” distributed across different time periods?
> >
> > We have added an analysis of return distribution over time in Appendix D.2, providing insights into how the "hyperportfolio" return evolves across different time periods.
> >
> >
> > > 4. For some of the time series, e.g. GDP as an economic time series, we would have existing forecasting beforehand. They might be part of the report, and the model can directly use these numbers, which are typically fairly accurate. In this case, how do you confirm that the agent utilizes the comprehensive information from different sources to make the “investment” instead of directly taking those numbers?
> >
> > We view the use of publicly available forecasts as a legitimate strategy, akin to what a human analyst might employ. In addition, the beforehand forecastings are not always reliable and can vary a lot even for famous models like GDPNow. For example, the Fed Banks’ GDP Estimates Vary Widely for the third quarter this year reported by Barron's. Thus, it is still a challenge to figure out the right prediction even provided with the forecasted values even for human experts, and requires the ability to integrate and verify diverse information sources.
> >
> >
> > > 5. Financial time series is typically the most noisy one compared with other time series, which should also be the least predictable. However, it is shown that the financial time series achieves the highest return. How do authors interpret such results?
> >
> > We delve into this in the newly added Appendix D. Firstly, it cannot be directly interpreted as a real financial return, as it does not fully follow the real market rules like trading hours, and we use daily close price instead of real-time data for public distribution as real-time data demands commercial access. However, the agent may have an information advantage since the close prices are delayed and do not reflect the latest news thus the agents are traded at a discounted price. We analyze the impact of using non-real-time data in Appendix D.1 by adjusting the financial returns with the minute-level data, which shows an overall decrement in the returns while still preserving the comparative advantages among agents. Secondly, the agent is still skewed by the information sources that frequently report big tech and medical companies, which leads to a portfolio similar to the case of the FANG+ index that tracks large trending companies. This strategy performs well in the experiment time span. We analyzed it in Appendix D.2.
> >
> >
> > > 6. Typos: equation 3.1.2, Table 3.2.1, and Figure 3.2.1 are not pointing to the correct place and are not indexed correctly. Table 1 in the Knowledge Base part should be Table 2.
> >
> > Thank you for pointing out these errors. We have corrected the mentioned typos and references, and have conducted a thorough review of the paper to ensure accuracy in all citations and references.
> >
> > ---
> >
> > We hope these responses adequately address your concerns and provide a clearer understanding of our work. We remain open to further queries and are committed to continually improving our manuscript. Thank you once again for your valuable feedback.

---

> > > ### Comment · Reviewer_apFV · 2023-11-22
> > >
> > > Dear Authors,
> > >
> > > I would like to thank you for your detailed answer. This has partially addressed my concerns. So I will raise my rating to reflect this. However, I still have some concerns regarding (1) if using existing forecasting is a legitimate strategy, then it's probably more reasonable to also include reliable existing forecasting as baselines. (2) It remains a bit unclear how long would the agent hold for each time series, which might shed some light on the decision of the agent. For example, this holding period may reflect the frequency of public forecasting. (3) The return of hyperportfolio as an evaluation metric of an agent is not easy to interpret, especially with so many time series combined together. A positive return does not necessarily mean that a wise decision is made. Take GDP as an example, since a lot of countries have positive growth, it's fairly easy for an agent to invest in such a time series and obtain a reasonable positive return.
> > >
> > > Sincerely,
> > >
> > > Reviewer apFV

---

> ### Author Response · Authors · 2023-11-22
>
> Dear Reviewer apFV,
>
> We sincerely appreciate your detailed feedback and your reconsideration of our work. Regarding your concerns:
>
> > (1) if using existing forecasting is a legitimate strategy, then it's probably more reasonable to also include reliable existing forecasting as baselines.
>
> We acknowledge there exist reliable domain-specific models in fields like quantitative economics and algorithmic trading. However, their focus differs from ours, our main focus is on general societal intelligence rather than domain-specific expertise. Such generality allows their applications to be extended beyond traditional time series forecasting such as business planning and social science research, as outlined in the introduction. Yet, directly competing with experts in all fields without domain-specific optimizations is not the target. We agree that comparison with domain experts can provide important insights into specific applications, but it will go beyond the main focus of this work.
>
> > (2) It remains a bit unclear how long would the agent hold for each time series, which might shed some light on the decision of the agent. For example, this holding period may reflect the frequency of public forecasting.
>
> The holding period for each time series is dynamic adjustments with multiple buy/sell operations in different amounts at different time points. Thus, there is no determined holding period for a time series. We recognize the value of visualizing these holding periods can provide useful insights. Considering the rebuttal period is limited, we will incorporate this into future revisions.
>
>
> > (3) The return of hyperportfolio as an evaluation metric of an agent is not easy to interpret, especially with so many time series combined together. A positive return does not necessarily mean that a wise decision is made. Take GDP as an example, since a lot of countries have positive growth, it's fairly easy for an agent to invest in such a time series and obtain a reasonable positive return.
>
> We have handled your concern on time series like GDP in **Section 3.2.2:** ***"Since some assets are almost ever-growing, such as GDP, which leads to risk-free returns, we wish all returns to come with risk, thus requiring good analysis to achieve a better risk-return trade-off."*** and further detailed in **Appendix C Robustness Analysis of SocioDojo**. We introduced mechanisms like overnight rate and explored the proper settings by experiment to eliminate "risk-free" returns for such time series to ensure every positive return can reflect an improvement in societal understanding and decision-making. We agree that developing more intuitive evaluation metrics for agents is an essential aspect of our future work, but also challenging akin to evaluating a fund in the real-world financial context.
>
> ---
>
> We hope this response can address your concerns and aid in the further evaluation of our work. Your feedback is invaluable, and we remain open to any further questions or suggestions you may have.
>
>
> Best regards,
>
> Authors of Submission 2968

---

### Author Response · Authors · 2023-11-20
**Response to All Reviewers**

We extend our sincere thanks to each reviewer for their insightful and constructive suggestions. In response to the feedback and concerns from reviewers, we have made significant updates to our manuscript, which are detailed below. These changes are highlighted in blue in the revised version:
1. We updated Table 3 with the Sharp ratio to measure risk-adjusted returns, we also updated the corresponding descriptions in Section 5.2.
2. We added Appendix B to introduce our methods in creating SocioDojo about how we reduce biases and ensure the reliability and diversity of sources.
3. We added Appendix D to analyze the impact of using daily close price data in finance instead of real-time data that is only accessible with commercial licenses and to study the distribution of returns over time.
4. We added Appendix E, an overview of the analyst-assistant-actuator architecture to provide an intuitive view of how agents process inputs in SocioDojo.
5. We fixed several typos.

---

### Meta-Review · Area_Chair_n3Ej · 2023-12-08

**Metareview:**

This paper presents the SocioDojo framework, designed for the development of autonomous agents capable of human-like analysis and decision-making in societal contexts, with a focus on areas like finance. The authors illustrate the application of SocioDojo through a task called "hyperportfolio", in which agents read news and analyze time series data to make decisions regarding asset buying and selling. The results demonstrate that the Analyst-Assistant-Actuator agent proposed in this study achieves significant improvements compared to the current state-of-the-art methods in both "standard" and "tracking" experimental settings.

The introduced environment and approach are innovative and highly valuable. I recommend the acceptance of this paper.

In line with common practices in NLP dataset papers, there are certain aspects highlighted by the reviewers that could be further refined:

1. Establishing a benchmark for human performance is crucial to provide context for the progress made in this work. Selecting representative expert performance can be a challenge, but it is essential for the community to gauge the advancements accurately. One potential approach, as suggested during discussions, could involve benchmarking against strategies employed in quantitative finance as a reference point.

2. It remains somewhat unclear to what extent the observed improvements can be attributed solely to the analytical capabilities of the proposed agent, as opposed to the potential influence of integrating human analysis reports. It would be beneficial to include experiments that involve running the agents on high-quality human analysis reports to distinguish the contributions of the agent's capabilities more clearly. This additional analysis would enhance the paper's comprehensibility and provide deeper insights into the agent's performance.

**Justification For Why Not Higher Score:**

- Lack of human performance that is crucial to provide context for the progress made in this work.
- It remains somewhat unclear to what extent the observed improvements can be attributed solely to the analytical capabilities of the proposed agent, as opposed to the potential influence of integrating human analysis reports.

**Justification For Why Not Lower Score:**

- The introduced environment is innovative and highly valuable.
- Good performance achieved by the proposed agent.

---

### Decision · Program_Chairs · 2024-01-16

Accept (spotlight)